# Trajectory-Based Neural Darwinism in Convolutional Neural Networks: Variation, Competition, and Selective Retention

## Abstract

Understanding how neural networks develop and stabilize internal representations remains a central challenge. Inspired by Edelman's Neural Darwinism, we introduce the Neuron Darwinian Dynamics System (NDDS), a trajectory-based framework that treats neurons as evolving agents under both local and global selective pressures. We define the Global Darwinian Pressure (GDP) as the population-average neuron fitness, capturing system-wide selection dynamics. Layer-wise analyses show that selective pressure intensifies over training, particularly in deeper layers, reflecting progressive consolidation of high-fitness neurons. Ablation experiments further reveal that removing survived neurons leads to substantial accuracy loss, whereas eliminating low-fitness neurons causes minimal degradation, demonstrating NDDS's ability to identify functionally critical units. Dynamic trajectory analyses show that survived neurons maintain coherent activity, stronger weights, and higher global Darwinian pressures, while eliminated neurons stagnate. Overall, our results support a Darwinian view of representation learning: networks achieve early-stage redundancy and later-stage specialization, enabling robust and stable task-relevant representations.

## 1 Introduction

The success of deep learning is often attributed to its ability to construct hierarchical feature representations Chizat & Netrapalli (2024); Banerjee (2025), yet the mechanisms that govern representational stability and neuron specialization remain only partially understood. Prior work has primarily emphasized optimization dynamics or information-theoretic principles Butakov (2024), while comparatively limited attention has been paid to competitive processes unfolding at the level of individual neurons. In neuroscience, Edelman's theory of Neural Darwinism proposes that neuronal populations evolve through variation, competition, and selective retention, thereby forming stable yet adaptable circuits Edelman (1987). Building on this perspective, we investigate whether analogous competitive dynamics emerge in artificial neural networks and how they shape robustness and specialization.

Motivated by this, we introduce a unified trajectory-based Darwinian framework—the Neuron Darwinian Dynamics System (NDDS)—which formalizes neuron evolution in convolutional architectures through the lens of survival and selection. NDDS integrates trajectory-based analyses of representational dynamics, layer-wise inspection of activations, weights, and embeddings over training, together with controlled ablation to rigorously quantify representational resilience. Within this framework, we define the Global Darwinian Pressure (GDP) as the population-average evolutionary fitness across all neurons, providing a scalar descriptor of the system-wide selective regime. Layer-wise analyses of GDP reveal that selective pressure systematically intensifies throughout training, with deeper layers exhibiting the largest gains, indicating progressive consolidation of high-fitness neurons across the hierarchy. Ablation studies highlight the functional importance of NDDS-selected neurons: removing survived neurons results in a substantial drop in test accuracy, whereas eliminating low-fitness neurons leads to only minor degradation, confirming that NDDS effectively identifies neurons critical for task performance. This integrated view enables systematic comparison of neuron-level dynamics across models of varying depth and dataset complexity. Our experimental evaluation covers a spectrum of architectures and datasets, beginning with a three-layer MLP on MNIST and progressively extending to ResNet-18 on CIFAR-10, VGG-16 on CIFAR-100, and ResNet-50 on Tiny-ImageNet. Across these

settings, neurons are categorized into survived, eliminated, and other groups according to long-term representational stability, providing a consistent lens for evaluating functional contributions. From an evolutionary perspective, the results reveal that different layers impose distinct selective pressures on neurons. Shallow layers exhibit highly variable and unstable trajectories, resembling an early exploration phase. Middle layers increasingly differentiate neurons into those maintaining sustained activity and those drifting toward quiescence, suggestive of emergent selective filtering. Deep layers show a tendency toward contraction, where a relatively compact subset of neurons retains higher activation while others decline. These observations are consistent with Darwinian dynamics of variation and selection, further supported by the ablation evidence of selective neuron importance. We restrict our analysis to Convolutional Neural Networks (CNNs) in this work, as their hierarchical structure and well-studied representational dynamics provide a controlled and interpretable setting for isolating neuron-level evolutionary mechanisms. In contrast, Transformers introduce attention-mediated interactions and layer normalization effects that confound neuron-level attribution, making them less suitable for our initial theoretical analysis. Collectively, these findings suggest that CNNs achieve robustness and representational specialization not solely through gradient-based optimization, but also through emergent neuron-level competition that parallels Darwinian selection principles.

## 2 Related Work

### 2.1 On Neural Networks Analysis

A large body of work has investigated how neural networks form and consolidate internal structure, spanning pruning, representational similarity, loss geometry, and interpretability. Pruning studies demonstrate that overparameterized models contain trainable sparse subnetworks, with the Lottery Ticket Hypothesis Frankle & Carbin (2019) and its extensions Liu (2019); Sanh (2020); Lee (2019); Evci (2020); Morcos (2019) showing that subnetworks can be identified via sensitivity measures Lee (2019), dynamic rewiring Evci (2020), or transfer across tasks Morcos (2019). Representation analyses such as SVCCA Raghu (2017) and CKA Kornblith (2019) reveal convergent layerwise structures, while neural tangent kernel theory Jacot (2018) and deep linear dynamics Saxe (2014) provide analytic descriptions of training. Complementing this analytic perspective, Geiger et al. Geiger et al. (2020) disentangle feature learning from lazy training, showing how trained networks can transition between kernel-like dynamics and representation-changing regimes. Geometric studies show low-loss mode connectivity Garipov (2018); Draxler (2018) and neural collapse phenomena Han (2022), connecting optimization to generalization. Interpretability methods including Network Dissection Bau (2017), TCAV Kim (2018), Integrated Gradients Sundararajan (2017), and SHAP Lundberg & Lee (2017) further expose concept-level features, while symmetry and re-basin analyses Ainsworth (2023) link parameter permutations to solution geometry. Finally, work on large-batch training Keskar (2017) and dynamical isometry Pennington (2017) elucidates how optimization biases shape solution quality. Taken collectively, these perspectives highlight redundancy, convergence, and selection-like pressures in neural networks, aligning with our Darwinian view of neuron-level competition.

### 2.2 Neuron Darwinian

The conceptual foundation for Darwinian mechanisms in neural systems was laid by Edelman's theory of neuronal group selection, which frames brain function as variation among neuronal populations, selective reinforcement of circuits, and inheritance of stable connectivity patterns Edelman (1987). Inspired by this paradigm, recent advances in artificial networks embed analogous variation–selection processes across computational scales, challenging the dominance of gradient-only optimization. Du et al. reinterpret late-epoch backprop-trained models as "ancestral genomes" and evolve offspring via differential evolution to reduce overfitting and accelerate inference Du (2024). At the neuron level, NeuroFS dynamically prunes and regrows inputs under synaptic-plasticity constraints to maintain adaptability under sparsity Zahra (2023). In dynamical systems, Czégel et al. show Darwinian neurodynamics in reservoir computing, where activity patterns are imperfectly copied and fitter variants selected, yielding emergent combinatorial problem solving Czégel (2021). Evolutionary processes also benefit spiking models: Shen et al. evolve excitatory–inhibitory circuits via spike-timing–dependent plasticity, achieving strong CIFAR-10 and ImageNet performance Shen (2023). At the architectural scale, Shafiee et al. encode heritable "DNA" for evolving compact offspring networks

Shafiee (2018), while Chen et al. propose OPNP, a gradient-sensitivity–based pruning scheme that improves out-of-distribution robustness by selecting fitter neurons and parameters Chen (2023). Collectively, these works demonstrate a convergent trend: embedding variation–selection mechanisms across synaptic, dynamical, and structural levels to improve adaptability, sparsity, and generalization beyond gradient descent. We extend this tracing with a neuron-level temporal analysis framework that tracks activation trajectories to distinguish "survived" from "eliminated" neurons, providing direct empirical evidence for Neural Darwinism in modern deep learning.

### 2.3 Neuron Trajectory

Recent work increasingly examines neuron trajectories—the evolution of individual activations or weights across layers and time—as a lens on training dynamics, interpretability, and generalization. Fu et al. formalize learning trajectories and derive generalization bounds tied to their complexity Fu (2023). Pesme and Flammarion analytically characterize gradient-flow paths in two-layer diagonal networks, showing convergence through successive saddles to minimal-norm solutions Pesme & Flammarion (2023), while Han et al. connect MSE training to the emergence of neural collapse by analyzing proximity and dynamics along the central path Han (2022), and Ahn links threshold-like neuron emergence to edge-of-stability dynamics Ahn (2023). In mechanistic interpretability, Conmy et al. introduce ACDC to extract activation subcircuits via trajectory-based graph discovery Conmy (2023), and Syed et al. apply attribution patching along activation paths to reveal causal transformer subcircuits Syed (2024). Beyond static analysis, Li et al. adapt trajectory forecasting (AMAG) to predict future neuron activity Li (2023), while spiking models leverage trajectory-inspired optimization to reduce firing load without loss of accuracy Shi (2024); Shen (2024). Together, these studies establish neuron trajectories as a unifying construct linking optimization dynamics, circuit discovery, and functional efficiency in modern networks.

## 3 Method

We formalize neuron evolution during training as a continuous-time dynamical system driven by both optimization gradients and intrinsic information-theoretic pressures. Intuitively, we treat each neuron as an evolving agent whose state is not only determined by its parameters but also by how it responds to data and gradients. This perspective allows us to study neural computation through the lens of dynamical systems and Darwinian selection Saxe (2014); Mei (2018); Chizat & Bach (2018). Importantly, we reserve the term "Darwinian" for neural dynamics that satisfy explicit, testable operational criteria involving variation, competition, selective retention, and non-circular prediction, as formalized in Section A.1.2.

Let a neural network $f_\theta : \mathcal{X} \to \mathcal{Y}$ consist of layers $\{L_k\}_{k=1}^D$, where layer $L_k$ contains neurons $\{a_i^{(k)}\}_{i=1}^{n_k}$. Each neuron is parameterized by a weight vector $w_i^{(k)} \in \mathbb{R}^{d_{k-1}}$, bias $b_i^{(k)} \in \mathbb{R}$, and activation function $\sigma$. Its activation at time $t$ is:

$$a_i^{(k)}(x,t) := \sigma\left(w_i^{(k)}(t)^\top h^{(k-1)}(x,t) + b_i^{(k)}(t)\right), \tag{1}$$

where $h^{(k-1)}$ is the output from $L_{k-1}$ and $h^{(0)} = x$. Thus, activations evolve jointly with weights and reflect both optimization and stochastic fluctuations Schoenholz (2017); Poole (2016).

### 3.1 Neuron Darwinian Dynamics System (NDDS)

**Definition 3.1** (Neuron State Vector). *To make this evolution explicit, we introduce the* neuron state vector, *which concatenates its trainable parameters, average activity, gradient statistics, and information-theoretic descriptors:*

$$\psi_i^{(k)}(t) := \left[w_i^{(k)}(t),\, b_i^{(k)}(t),\, \mu_i^{(k)}(t),\, g_i^{(k)}(t),\, \mathcal{I}_i^{(k)}(t)\right]. \tag{2}$$

Here we explicitly define each component and its domain/estimation modality:

$$\mu_i^{(k)}(t) := \mathbb{E}_{x \sim \mathcal{D}}\big[a_i^{(k)}(x, t)\big], \tag{3}$$

$$g_i^{(k)}(t) := \mathbb{E}_{x \sim \mathcal{D}}\bigg[\frac{\partial \mathcal{L}(x)}{\partial a_i^{(k)}(x, t)}\bigg], \tag{4}$$

$$\mathcal{I}_i^{(k)}(t) := H\Big(a_i^{(k)}(\cdot, t)\Big) = -\int p_i^{(k)}(a, t) \log p_i^{(k)}(a, t) \, da, \tag{5}$$

Here $p_i^{(k)}(a, t)$ denotes the marginal distribution of the neuron activation induced by $x \sim \mathcal{D}$.

We emphasize estimation modality: expectations are taken with respect to the data distribution $\mathcal{D}$; in practice they are approximated by empirical estimates over mini-batches. Throughout we reserve the symbol $\mathcal{L}(x)$ to denote the per-example loss.

The evolution of each neuron is then modeled as a differential equation:

$$\frac{d}{dt}\psi_i^{(k)}(t) = \mathbf{F}_\theta^{(k)}\big(\psi_i^{(k)}(t), \mathcal{D}, \mathcal{L}\big), \tag{6}$$

where $\mathbf{F}_\theta^{(k)}$ captures the joint effect of gradient-descent-like updates and intrinsic representational dynamics. This abstraction allows us to borrow tools from dynamical systems theory to analyze stability, convergence, and diversity of neurons Achille & Soatto (2018b). The NDDS itself is therefore a descriptive dynamical-system model. A network is claimed to exhibit Darwinian neural dynamics when the additional operational criteria in Definition A.1 are satisfied.

**Assumption 3.2** (Smooth and Bounded Dynamics). *We adopt a hypothesis compatible with practical discrete optimization. The parameter trajectory $\theta(t)$ is assumed to be absolutely continuous and piecewise $C^1$ in $t$ (so that it admits a time-continuous interpolation), and $\mathbf{F}_\theta^{(k)}$ is locally Lipschitz in $\psi$ on trajectories of interest. This formulation explicitly permits discretization effects arising from SGD and non-smooth activations (e.g. ReLU) by interpreting derivatives in the sense of absolutely continuous interpolation or Clarke subgradients when necessary. We assume standard smoothness and boundedness conditions on interpolated trajectories; detailed assumptions and discretization–continuum error bounds are deferred to Appendix.*

**Assumption 3.3** (Local Gaussianity of pre-activations and diagnostic protocol). *To avoid conflicts with non-negative, mass-at-zero activations (e.g. ReLU), we state the main parametric approximation at the pre-activation level. Define the pre-activation*

$$z_i^{(k)}(x, t) := w_i^{(k)}(t)^\top h^{(k-1)}(x, t) + b_i^{(k)}(t), \tag{7}$$

*and its smoothed version*

$$\tilde{z}_i^{(k)}(x, t) := z_i^{(k)}(x, t) + \varepsilon, \qquad \varepsilon \sim \mathcal{N}(0, \sigma_\varepsilon^2). \tag{8}$$

*Diagnostic procedures, variance proxies, and fallback strategies are deferred to the Appendix.*

### 3.2 Trajectory-Based Evolutionary Fitness

**Definition 3.4** (Neuron Trajectory). *The trajectory of a neuron in state space is defined as*

$$\Gamma_i^{(k)} := \{\psi_i^{(k)}(t) \mid t \in [0, T]\}. \tag{9}$$

From this path we extract three complementary quantities:

**Definition 3.5** (Trajectory Length). *The trajectory length of neuron $i$ in layer $k$ is the cumulative representational movement of its state vector $\psi_i^{(k)}$ measured with a block-wise scaling matrix $D^{(k)}$ that normalizes heterogeneous components of $\psi$:*

$$\mathcal{A}_i^{(k)} := \int_0^T \Big\| D^{(k)} \frac{d\psi_i^{(k)}(t)}{dt} \Big\|_2 \, dt, \tag{10}$$

where $D^{(k)}$ is taken to be block-diagonal with positive diagonal blocks that rescale each block of $\psi$. The block-wise construction ensures no single block systematically dominates the norm and makes the quantity invariant to simple coordinate scalings within each block. For comparability across different training durations we use a time-averaged trajectory length; its formal definition and the discrete approximations used in experiments appear in the Appendix.

**Definition 3.6** (Integrated Entropy). *The integrated entropy of neuron i accumulates a per-time estimate of the neuron's entropy over training:*

$$\mathfrak{H}_i^{(k)} := \int_0^T \mathcal{I}_i^{(k)}(t)\,dt, \tag{11}$$

where $\mathcal{I}_i^{(k)}(t)$ denotes a numerically stable estimator of the neuron's differential entropy at time $t$. For comparability we also consider the time-averaged form $\overline{\mathfrak{H}}_i^{(k)} := \frac{1}{T}\mathfrak{H}_i^{(k)}$.

When the Gaussian plug-in is appropriate we use the variance-proxy with explicit numerical stabilization:

$$\tilde{\mathcal{I}}_i^{(k)}(t) := \tfrac{1}{2}\log\big(\mathrm{Var}_x[z_i^{(k)}(x,t)] + \sigma_\varepsilon^2 + \epsilon_{\mathrm{var}}\big), \tag{12}$$

where $\sigma_\varepsilon^2 > 0$ is the additive smoothing noise variance introduced in Assumption 3.3 and $\epsilon_{\mathrm{var}} > 0$ is a small numeric floor (e.g. $10^{-8}$) to avoid $\log(0)$ and ensure robust estimation in finite samples. Note that the Gaussian plug-in differs from the differential entropy by the additive constant $\frac{1}{2}\log(2\pi e)$; when absolute entropy values are needed this constant is accounted for in post-processing.

**Definition 3.7** (Ablation-based Utility). *For neuron i in layer k define the instantaneous ablation-based utility*

$$U_i^{(k)}(t) := \mathbb{E}_{x\sim\mathcal{D}}\big[\mathcal{L}(f_{\theta(t)\setminus i};x) - \mathcal{L}(f_{\theta(t)};x)\big], \tag{13}$$

where $f_{\theta\setminus i}$ denotes the network obtained by zeroing neuron $i$'s activation. By this convention $U_i^{(k)}(t) > 0$ indicates the neuron is useful at time $t$.

**Definition 3.8** (Time-averaged Utility). *For comparability across training durations we use the time-averaged utility*

$$\overline{U}_i^{(k)} := \frac{1}{T}\int_0^T U_i^{(k)}(t)\,dt. \tag{14}$$

**Definition 3.9** (Evolutionary Fitness). *To ensure comparability across heterogeneous quantities we first perform layer-wise standardization (z-scoring) of each constituent statistic. Here $\mathrm{SD}_j(\cdot)$ denotes the sample standard deviation over neurons j in layer k. Let*

$$\hat{\overline{U}}_i^{(k)} := \frac{\overline{U}_i^{(k)} - \mathbb{E}_j[\overline{U}_j^{(k)}]}{\mathrm{SD}_j(\overline{U}_j^{(k)})},$$

$$\hat{\overline{\mathcal{A}}}_i^{(k)} := \frac{\overline{\mathcal{A}}_i^{(k)} - \mathbb{E}_j[\overline{\mathcal{A}}_j^{(k)}]}{\mathrm{SD}_j(\overline{\mathcal{A}}_j^{(k)})}, \tag{15}$$

$$\hat{\overline{\mathfrak{H}}}_i^{(k)} := \frac{\overline{\mathfrak{H}}_i^{(k)} - \mathbb{E}_j[\overline{\mathfrak{H}}_j^{(k)}]}{\mathrm{SD}_j(\overline{\mathfrak{H}}_j^{(k)})}.$$

The evolutionary fitness is the convex combination

$$\Phi_i^{(k)} := \alpha\,\hat{\overline{U}}_i^{(k)} + \beta\,\hat{\overline{\mathcal{A}}}_i^{(k)} + \gamma\,\hat{\overline{\mathfrak{H}}}_i^{(k)}, \qquad \alpha,\beta,\gamma > 0, \tag{16}$$

where the weights $\alpha,\beta,\gamma$ are chosen from a small recommended grid after layer-wise normalization or selected via a held-out validation objective. *The held-out split used to tune $\alpha,\beta,\gamma$ is kept separate from the held-out split used for the non-circular prediction test in Section A.1.2.*

To relate neuron-level Darwinian quantities to a system-wide descriptor, we introduce an aggregate index that summarizes the global selective pressure emerging from the NDDS. This quantity plays an analogous role to a population-level fitness landscape in evolutionary dynamics and provides a scalar diagnostic of the network's collective representational evolution.

**Definition 3.10** (Global Darwinian Pressure). *Let $\Phi_i^{(k)}$ denote the standardized evolutionary fitness of neuron i in layer k as defined in equation 16. The* Global Darwinian Pressure (GDP) *at training horizon T is defined as the population average of neuron-level fitness:*

$$\mathrm{GDP}(T) \; := \; \frac{1}{N} \sum_{k=1}^{D} \sum_{i=1}^{n_k} \Phi_i^{(k)}(T), \tag{17}$$

*where $N = \sum_{k=1}^{D} n_k$ is the total number of neurons.*

### 3.3 Selection and Survival Criteria

To connect neuron-level fitness with both local and system-wide selective forces, we couple layer-wise survival thresholds with the Global Darwinian Pressure (GDP) introduced in Definition 3.10. Whereas $\Phi_i^{(k)}$ reflects an individual neuron's evolutionary fitness under NDDS dynamics, the GDP provides a scalar descriptor of the population's global selective regime. Survival is therefore determined not only relative to peers within the same layer, but also relative to the network-wide evolutionary baseline.

**Definition 3.11** (Survived Neuron). *Neuron i in layer k is said to be* survived *if its standardized fitness exceeds both a layer-local statistical threshold and the system-wide evolutionary baseline:*

$$\begin{aligned}
\Phi_i^{(k)}(T) &\geq \mathbb{E}_j[\Phi_j^{(k)}(T)] + \lambda \, \mathrm{SD}(\Phi_j^{(k)}(T)), \\
\Phi_i^{(k)}(T) &\geq \mathrm{GDP}(T), \qquad \lambda > 0.
\end{aligned} \tag{18}$$

The first condition enforces *intra-layer* selection, ensuring that only neurons whose representational behavior is significantly above their local peers persist. The second condition imposes a global evolutionary constraint: a neuron must also outperform the aggregate population-level fitness landscape. Together, these two pressures instantiate an NDDS-specific analogue of multi-scale evolutionary selection, where neurons compete both within layers and against the global ecological niche shaped by the GDP Han (2015); Frankle & Carbin (2019); Morcos (2019).

**Definition 3.12** (Eliminated Neuron). *A neuron i in layer k is said to be* eliminated *at training horizon T if its standardized evolutionary fitness falls significantly below its layer-local distribution, namely*

$$\Phi_i^{(k)}(T) \; < \; \mathbb{E}_j[\Phi_j^{(k)}(T)] \; - \; \lambda \, \mathrm{SD}(\Phi_j^{(k)}(T)), \qquad \lambda > 0. \tag{19}$$

*This criterion designates neurons whose representational contribution is consistently substandard relative to peers in the same layer.*

**Definition 3.13** (Other Neuron). *A neuron i in layer k is classified as an* other neuron *if it satisfies neither the survival condition in Definition 3.11 nor the elimination condition in Definition 3.12. Formally,*

$$i \in \text{Other} \quad \Longleftrightarrow \quad i \notin \text{Survived} \;\; \text{and} \;\; i \notin \text{Eliminated}. \tag{20}$$

**Lemma 3.14** (Instability with sustained entropy decay implies vanishing fitness). *Let the evolutionary fitness $\Phi_i^{(k)}(T)$ be defined as in equation 16, namely*

$$\Phi_i^{(k)}(T) = \alpha \, \widehat{\overline{U}}_i^{(k)}(T) + \beta \, \widehat{\overline{\mathcal{A}}}_i^{(k)}(T) + \gamma \, \widehat{\overline{\mathfrak{H}}}_i^{(k)}(T), \qquad \alpha, \beta, \gamma > 0. \tag{21}$$

*Assume there exist constants $c_H > 0$, $C_H < \infty$, and $T_0 \geq 0$ such that for all $T \geq T_0$ the standardized time-averaged entropy satisfies*

$$\widehat{\overline{\mathfrak{H}}}_i^{(k)}(T) \leq -c_H T + C_H. \tag{22}$$

*Assume further that the utility and trajectory-adaptation terms do not asymptotically compensate for this entropy collapse, namely*

$$\overline{\hat{U}}_i^{(k)}(T) = o(T), \qquad \overline{\hat{\mathcal{A}}}_i^{(k)}(T) = o(T). \tag{23}$$

*Then for any fixed positive weights $\alpha, \beta, \gamma > 0$ in equation 16, we have*

$$\lim_{T \to \infty} \Phi_i^{(k)}(T) = -\infty. \tag{24}$$

**Definition 3.15** (Gradient–Variance Contribution). *For a neuron i in layer k we define the instantaneous gradient second moment*

$$q_i^{(k)}(t) := \mathbb{E}_x \left[ \left( \frac{\partial \mathcal{L}(x)}{\partial a_i^{(k)}(x, t)} \right)^2 \right], \tag{25}$$

*and the instantaneous activation variance*

$$\sigma_i^{2(k)}(t) := \operatorname{Var}_x \left[ a_i^{(k)}(x, t) \right]. \tag{26}$$

*We then define the (time-averaged)* gradient–variance contribution *by*

$$\Delta_i^{(k)} := \frac{1}{T} \int_0^T \mathbb{E}_x \left[ \left( \frac{\partial \mathcal{L}(x)}{\partial a_i^{(k)}(x, t)} \right)^2 \cdot \operatorname{Var}_x \left[ a_i^{(k)}(x, t) \right] \right] dt \tag{27}$$

**Theorem 3.16** (Fitness Threshold Implies Gradient–Variance Contribution). *Let*

$$\Delta_i^{(k)} := \frac{1}{T} \int_0^T q_i^{(k)}(t) \, \sigma_i^{2(k)}(t) \, dt, \tag{28}$$

*where*

$$q_i^{(k)}(t) := \mathbb{E}_x \left[ \left( \frac{\partial \mathcal{L}(x)}{\partial a_i^{(k)}(x, t)} \right)^2 \right], \qquad \sigma_i^{2(k)}(t) := \operatorname{Var}_x \left[ a_i^{(k)}(x, t) \right]. \tag{29}$$

*Suppose Assumptions 3.2 and 3.3 hold. Assume that the gradient second moment is uniformly non-vanishing, i.e., there exists $\underline{c}_g > 0$ such that*

$$q_i^{(k)}(t) \geq \underline{c}_g \qquad \textit{for almost every } t \in [0, T]. \tag{30}$$

*Assume further that high fitness is entropy-supported: there exist constants $\tau > 0$ and $v_\tau > 0$ such that*

$$\Phi_i^{(k)}(T) \geq \tau \quad \implies \quad \frac{1}{T} \int_0^T \sigma_i^{2(k)}(t) \, dt \geq v_\tau. \tag{31}$$

*Then*

$$\Phi_i^{(k)}(T) \geq \tau \quad \implies \quad \Delta_i^{(k)} \geq \kappa, \tag{32}$$

*where*

$$\kappa := \underline{c}_g v_\tau > 0. \tag{33}$$

This result bridges our tracing-based measure with a classical signal-to-noise criterion under an entropy-supported high-fitness regime. Since the revised fitness assigns a positive contribution to trajectory adaptation, high fitness should not be interpreted as arising solely from gradient–variance effects. Rather, when high fitness is accompanied by non-collapsed activation variance, the neuron necessarily contributes to non-trivial gradient–variance interactions Achille & Soatto (2018a); Martens (2020). Under the NDDS, neurons are treated as evolving agents competing under both local and global Darwinian pressures, and the GDP provides a compact scalar descriptor of the network's emergent ecological landscape.

Overall, the NDDS provides a principled framework to study representational dynamics under Neural Darwinism. By coupling neuron-level evolutionary fitness with the system-wide GDP, the framework treats neurons not as static units with fixed importance, but as evolving entities embedded in a multi-scale selective environment. Neurons compete for survival through their trajectory length, stability, entropy, and their ability to surpass both local and global evolutionary baselines. This formalism explains empirical neuron-pruning phenomena, predicts layer-to-layer propagation of specialization, and reveals how global selective forces shape the emergence of structured internal representations Raghu (2017); Jacot (2018).

# 4    Experiments

We designed a series of experiments to determine whether convolutional neural networks exhibit dynamics consistent with Neural Darwinism and to understand how such processes emerge across different layers and architectures. Our analysis begins with layer-wise studies. These analyses allow us to examine how selective pressure varies across the hierarchy. Building on these insights, we then conduct controlled ablation experiments—starting with a CNN trained on MNIST—to quantitatively assess representational resilience by selectively removing survived or eliminated neurons identified by NDDS. Finally, we extend our trajectory-based evaluation to larger architectures and datasets, focusing primarily on ResNet-50 trained on Tiny-ImageNet, with additional experiments on a three-layer MLP-Net (MNIST), ResNet-18 (CIFAR-10), and VGG-16 (CIFAR-100) included in the Appendix. Across all settings, neurons are categorized into survived, eliminated, and other groups based on long-term representational stability, providing a unified comparative framework across layers, architectures, and scales.

## 4.1    Selection Dynamics of Global Darwinian Pressure Across Network Layers

In Figure 1, we perform a layer-wise analysis of Global Darwinian Pressure (GDP) in a three-layer multilayer perceptron trained on MNIST. The left panel compares the mean GDP at Epoch 10 and Epoch 50, with error bars reporting variability across repeated measurements. The results reveal a depth-dependent redistribution of selective pressure during training. Specifically, the first layer shows a reduction in GDP, whereas the second and third layers exhibit increased GDP, with the largest relative gain occurring in the deepest layer. This indicates that selective pressure is not uniformly amplified across the hierarchy; instead, shallow layers appear to stabilize after early feature formation, while deeper layers undergo stronger selection-driven consolidation as task-relevant representations emerge. The percentage-change analysis in the right panel further quantifies this layer-dependent effect. The negative change in the first layer contrasts with the positive changes in the deeper layers, showing that GDP evolution is organized by network depth rather than by a global monotonic shift. These results support our interpretation that Darwinian pressure becomes increasingly concentrated in later layers, where neurons are more directly involved in class-discriminative representation and selective retention.

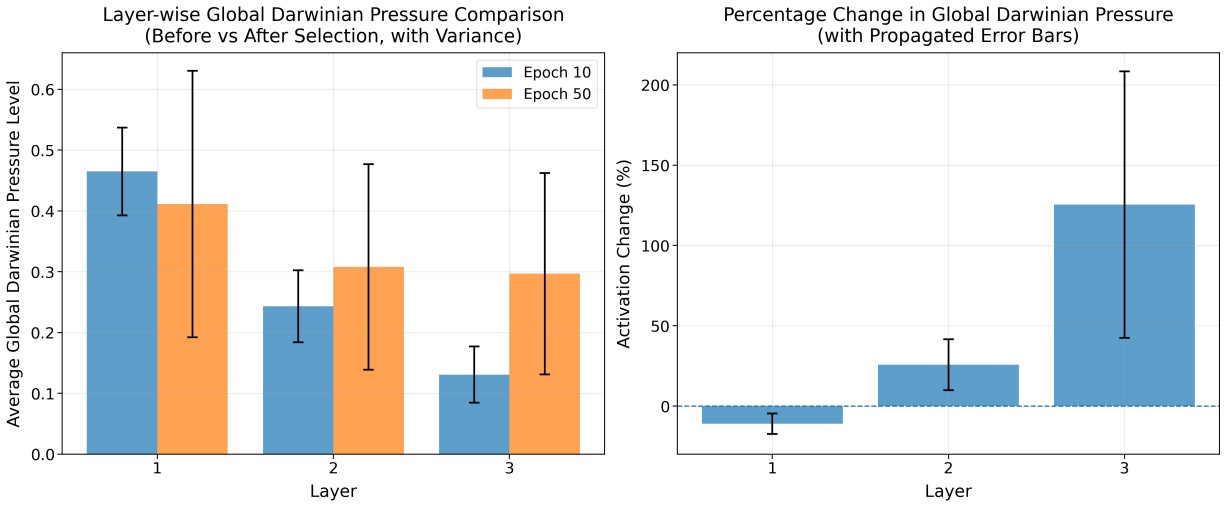

Figure 1: Layer-wise Global Darwinian Pressure Changes Under the Neuron Darwinian Dynamics System.

## 4.2    Ablation Experiment

### 4.2.1    Ablation Experiment with Random Neuron Removal

In Figure 2, we conducted ablation experiments on a CNN trained on MNIST to evaluate the resilience of its internal representations under progressive neuron removal. In the unperturbed network, accuracy reaches

99.3%, and the t-SNE projection shows tight, well-separated digit clusters, indicating a highly structured latent space. With 30% of neurons ablated, accuracy remains 99.0%, and cluster compactness and separation are largely preserved, suggesting substantial representational redundancy. At 60% ablation, accuracy decreases to 98.3%, and clusters begin to expand and partially overlap, though the global structure persists, implying that representational burden is reallocated to the remaining neurons. A qualitatively different pattern appears at 90% ablation: accuracy collapses to 64.9%, and the t-SNE embedding loses all cluster structure. Overall, these results support a Darwinian view of neural representations: moderate ablation removes redundant neurons while preserving accuracy and geometric separability, but once ablation encroaches on the stabilized subset critical for task-relevant structure, both accuracy and representation quality collapse.

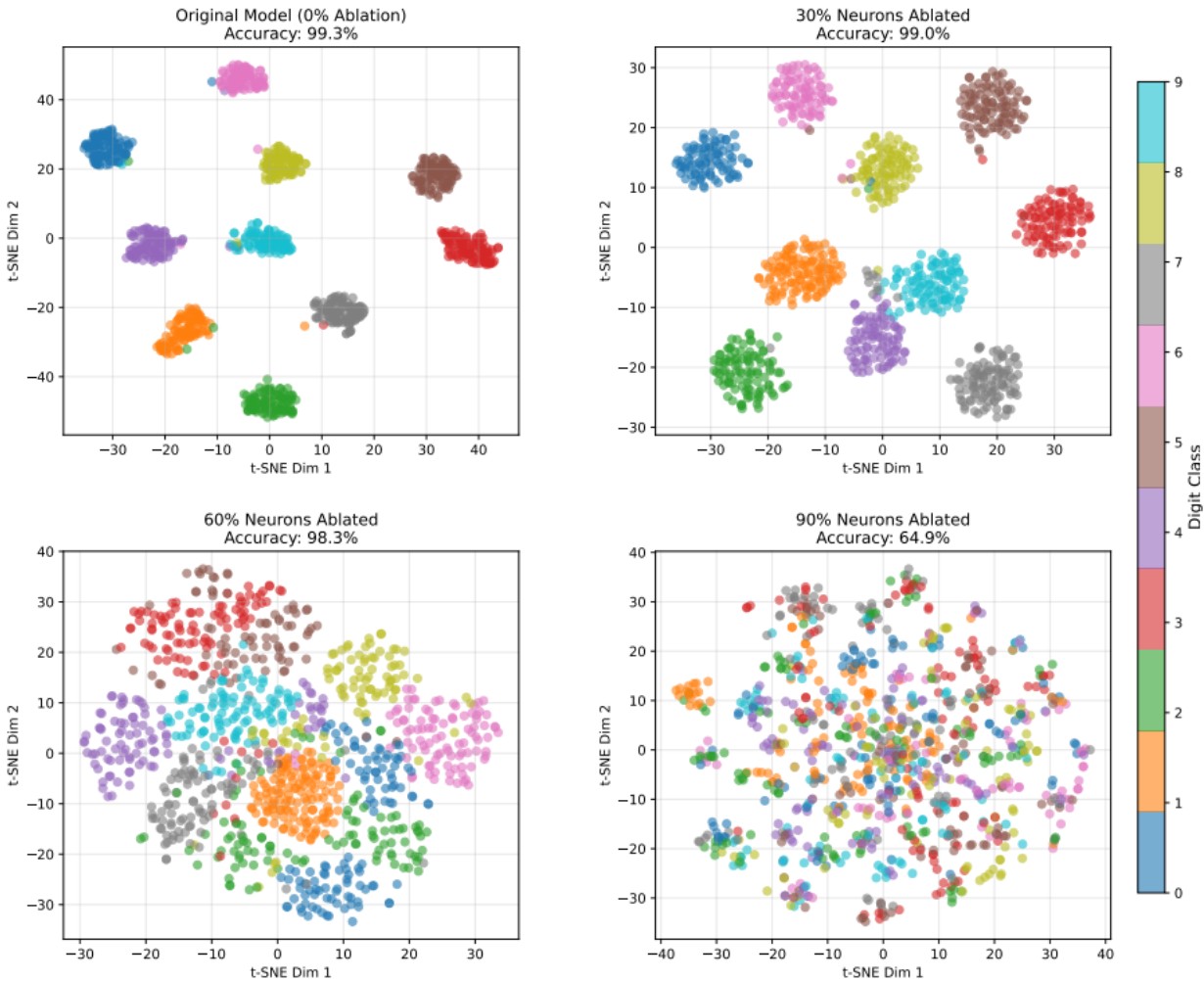

Figure 2: Ablation Experiment on MNIST with Random Neuron Removal.

### 4.2.2 Ablation Experiment with Neuron Darwinian Dynamics System

In Figure 3, we evaluated the effectiveness of the Neuron Darwinian Dynamics System in identifying functionally critical components within the network by applying its Global Darwinian Pressure mechanism to categorize neurons into survived and eliminated groups. Using MNIST as the benchmark task, the baseline model achieved 98.30% accuracy. After ablating the neurons marked as survived under Global Darwinian Pressure, model accuracy dropped markedly to 88.77%, indicating that the evolutionary selection pressure successfully isolated neurons essential for maintaining high discriminative performance. In contrast, removing the neurons categorized as eliminated resulted in only a minor reduction to 96.36%, suggesting that these units contribute redundant or non-critical features. These findings demonstrate that the Darwinian selection

framework effectively distinguishes neurons with high functional importance from those that can be pruned with minimal performance impact.

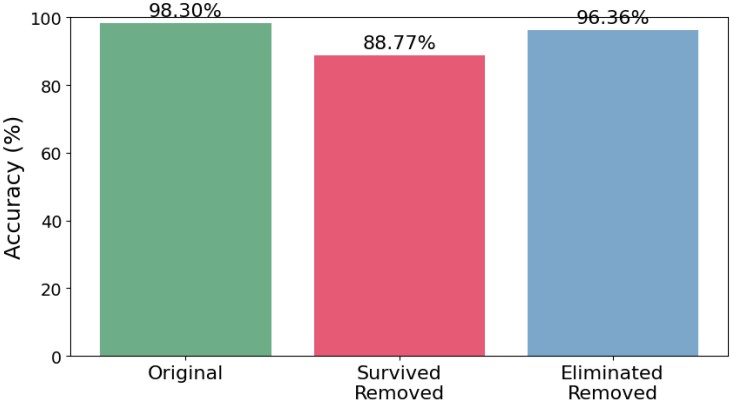

Figure 3: Ablation Results Under Global Darwinian Pressure on MNIST with Neuron Darwinian Dynamics System.

### 4.3 ResNet-50 on Tiny-ImageNet

### 4.3.1 Dynamics Neuron Trajectory and Evolution Analysis

The dynamic PCA trajectories for the shallow layer (Figure 4(a), top) reveal temporal representational changes across training. Each trajectory reflects the evolution of a neuron's activation statistics in a low-dimensional space. Survived neurons generally trace longer, more directionally consistent paths, indicating representational refinement and stronger task-related adaptation. Their trajectories drift toward structured regions of the PCA manifold, suggesting non-random reorganization that supports discriminative feature encoding. In contrast, eliminated neurons follow shorter, less exploratory paths that remain close to initialization, consistent with representational stagnation expected under early-stage selective pruning. Quantitatively, by the final epoch (Figure 4(c), top), survivors reach a median trajectory length of 3.2 units, compared to 2.4 for eliminated neurons and 2.3 for the other group. This indicates that sustained representational movement—rather than initial position—associates with retention. Weight magnitude evolution (Figure 4(d), top) shows only minor group differences: eliminated neurons maintain slightly higher L2 norms, with the other group lowest. Overall stability suggests that in shallow layers, synaptic resource allocation remains relatively fixed, with large-scale reallocation not yet evident.

Middle-layer PCA trajectories (Figure 4(a), middle) exhibit stronger divergence across neuron types. Survived neurons traverse extended, often curved paths primarily along PC1 (96.7% variance explained), with modest PC2 modulation. Clustering is weak; dispersion dominates. Eliminated neurons show substantially shorter displacements, remaining near initialization with fragmented paths. The other group exhibits moderate movement but not the sustained displacement of survivors. Trajectory length evolution (Figure 4(c), middle) highlights this separation: by training's end, survivors reach 3.8 cumulative units, eliminated neurons 2.8, and others even lower. This wider gap underscores increasing importance of sustained representational plasticity at mid-level stages. Weight magnitude evolution (Figure 4(d), middle) shows stable rankings: eliminated neurons hold slightly higher norms than survivors. Their stronger initial parameterization was not matched by functional adaptation.

Deep-layer trajectories (Figure 4(a), bottom) reveal the strongest differentiation in representational mobility. Survived neurons follow long, structured arcs, reflecting continued refinement of high-level semantic representations and convergence toward a compact PCA subregion, consistent with emerging attractor-like states. Eliminated neurons show minimal displacement beyond early epochs, indicating rapid stagnation. Other neurons display partial mobility but lack sustained directional movement. Trajectory lengths (Figure 4(c), bottom) accentuate this: survivors reach 7 units, eliminated neurons 4. This large gap shows

strong association between deep-layer survival and representational plasticity. Weight magnitude evolution (Figure 4(d), bottom) exhibits global decay across all neuron types, with survivors and eliminated neurons following similar trajectories and only slight divergence at convergence; the other group trends lower. These results suggest reduced differentiation of synaptic strength in deeper layers, with survival linked to only marginally higher residual weights. Overall, selection pressure strengthens with depth: shallow layers show subtle differences, middle layers display intensified divergence driven by sustained plasticity, and deep layers exhibit consolidation—neurons undergoing larger representational changes are more likely to be retained. These patterns align with Neural Darwinism principles of variation, competition, and selective retention.

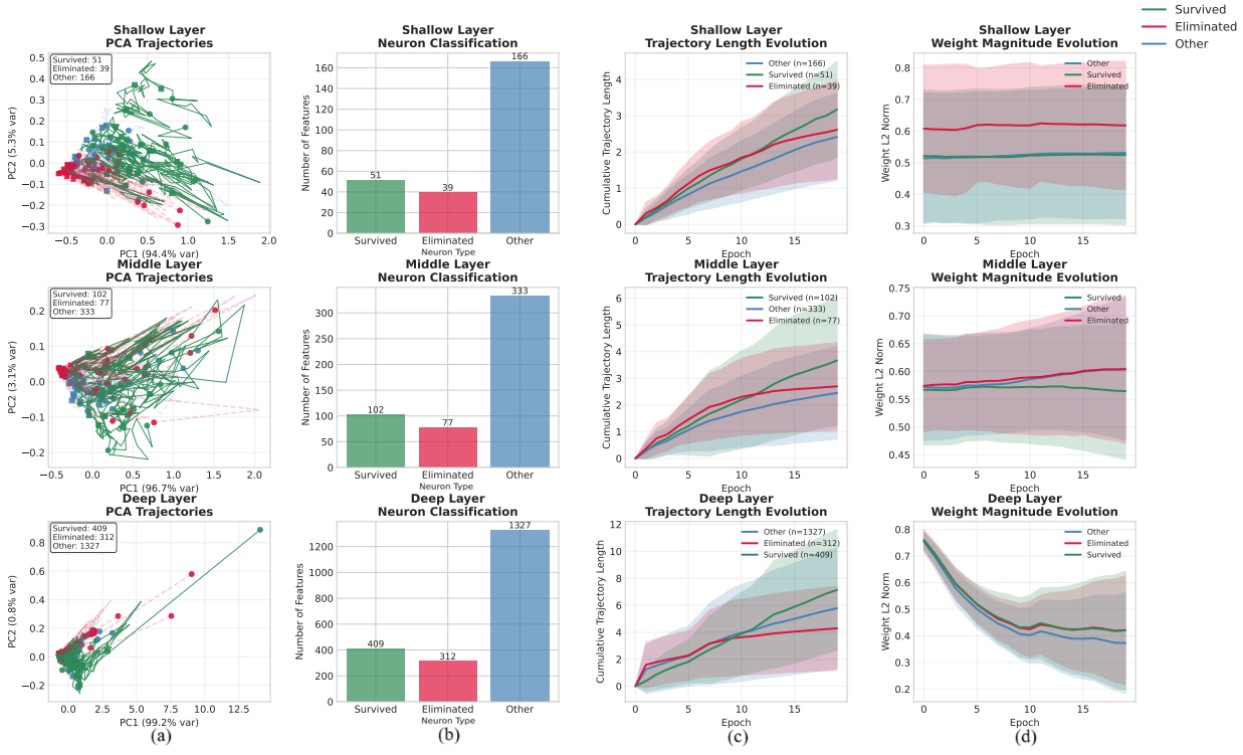

Figure 4: Dynamics Neuron Trajectory and Evolution Analysis on Tiny-ImageNet. Shaded regions represent the standard deviation of the corresponding quantity.

### 4.3.2 Static PCA and Global Darwinian Pressure Evolution

Figure 5 shows static PCA projections of final neuron states (top) and mean global Darwinian pressure evolution (bottom) across shallow, middle, and deep layers. In the shallow layer, PC1 explains 94.4% of the variance (PC2 5.3%), indicating that the final population is largely confined to a single dominant axis. Surviving neurons (green) occupy a moderately dispersed region displaced from the origin, consistent with coordinated stabilization without collapse into a tight cluster. Eliminated neurons (red) form a compact cluster near the lower-left quadrant, reflecting low global Darwinian pressure. Other neurons (blue) lie between these groups. Pressure dynamics match this structure: survivors maintain higher, stable norms; eliminated neurons decay toward near-zero; others follow mixed trajectories.

In the middle layer PC1 explains 96.7% of variance (PC2 3.1%), showing even stronger alignment to a dominant axis. Survivors lie in the central-to-positive PC1 range, eliminated neurons at the negative extreme, and other neurons in between. Their pressure evolutions mirror this ordering: survivors remain high, eliminated neurons rapidly decay, and others decline moderately.

In the deep layer PC1 captures 99.2% of variance (PC2 0.8%), indicating near-complete ordering along one axis. Neurons cluster densely, with eliminated units at the low-PC1 boundary and survivors extending

toward the positive tail. Pressure dynamics show early rises and stabilization for survivors, rapid decay for eliminated neurons, and mid-range plateaus for intermediates. Overall, depth intensifies selection-like dynamics: growing dominance of a single principal axis and widening divergence in pressure trajectories yield increasingly specialized representations.

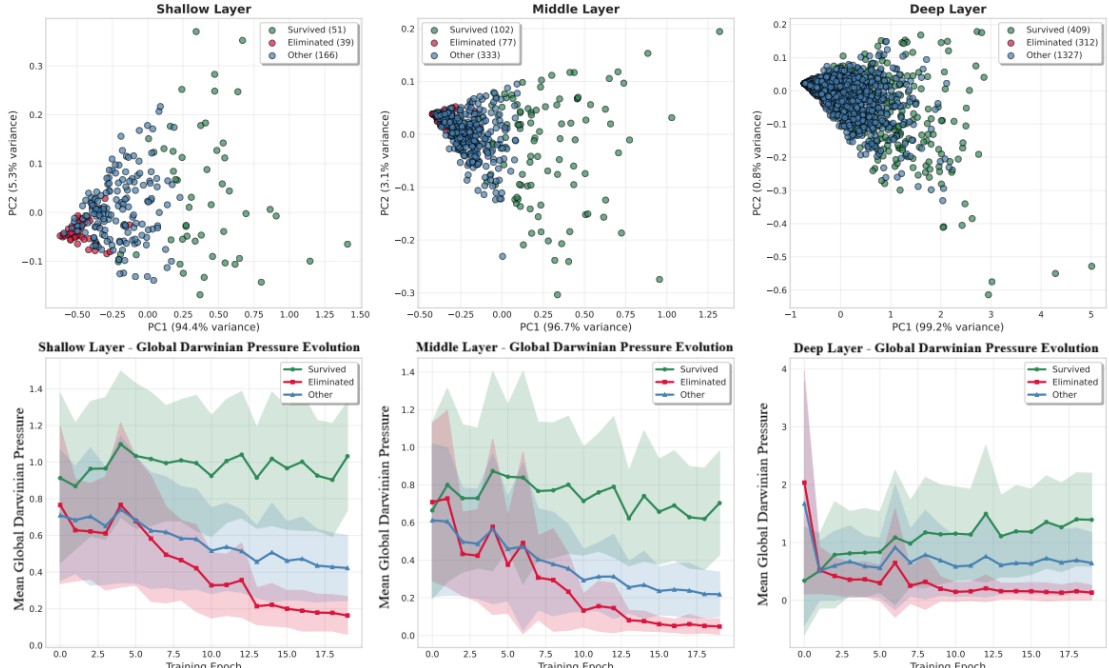

Figure 5: Static PCA and Global Darwinian Pressure Evolution on Tiny-ImageNet. Shaded regions represent the standard deviation computed across neurons of the same category.

## 5 Conclusion

In this work, we introduced the Neuron Darwinian Dynamics System (NDDS), a unified trajectory-based framework for characterizing neuron-level evolution in convolutional architectures. By defining neuron fitness and the Global Darwinian Pressure (GDP), NDDS provides a principled way to quantify system-wide selective forces and to track how representational competition unfolds during training. Across all evaluated models and datasets, we consistently observe hallmark signatures of Darwinian dynamics: neurons begin with highly diverse trajectories, but only a subset maintains coherent activity patterns, stronger and more stable weights, and persistently high global Darwinian pressures. The remaining neurons drift toward representational stagnation and are effectively eliminated within the population dynamics. Layer-wise analyses reveal that selective pressure intensifies over the course of training, particularly in deeper layers, which progressively consolidate a compact group of high-fitness neurons that dominate task-relevant encoding. Ablation studies further validate the functional meaning of these dynamics: removing survived high-fitness neurons leads to substantial degradation in accuracy, while eliminating low-fitness neurons has minimal effect. This demonstrates that NDDS not only captures representational evolution but also identifies units essential for the model's final performance and robustness.

Overall, our findings support a Darwinian perspective on representation learning in CNNs: optimization produces early-stage redundancy followed by later-stage specialization driven by emergent competitive pressures among neurons. This dual view—combining gradient-based learning with selection-like population dynamics—offers a complementary explanation for how deep networks achieve stable, high-quality representations. Future work may extend NDDS to architectures such as recurrent networks and Transformers, investigate how attention mechanisms reshape neuron-level competition, and explore implications for pruning, interpretability, and biologically grounded computational models.

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

# A    Appendix

## A.1    Further Methodological Details

### A.1.1    Notation and Preliminaries

To maintain consistency with the main text, we briefly recap key notations:

- Neural network $f_\theta : \mathcal{X} \to \mathcal{Y}$, layers $\{L_k\}_{k=1}^D$, where layer $k$ contains $n_k$ neurons indexed by $i$.

- Parameters of neuron $i$ at layer $k$: weights $w_i^{(k)}(t) \in \mathbb{R}^{d_{k-1}}$, bias $b_i^{(k)}(t) \in \mathbb{R}$, activation function $\sigma$.

- Activation:
$$a_i^{(k)}(x,t) := \sigma\left(\langle w_i^{(k)}(t), h^{(k-1)}(x,t)\rangle + b_i^{(k)}(t)\right). \tag{34}$$

- Neuron state vector (compound state):
$$\psi_i^{(k)}(t) := \left[w_i^{(k)}(t),\, b_i^{(k)}(t),\, \mu_i^{(k)}(t),\, g_i^{(k)}(t),\, \mathcal{I}_i^{(k)}(t)\right], \tag{35}$$
where
$$\mu_i^{(k)}(t) = \mathbb{E}_{x \sim \mathcal{D}}[a_i^{(k)}(x,t)], \quad g_i^{(k)}(t) = \mathbb{E}_{x \sim \mathcal{D}}\left[\frac{\partial \mathcal{L}(x)}{\partial a_i^{(k)}(x,t)}\right], \tag{36}$$
and $\mathcal{I}_i^{(k)}(t)$ is the instantaneous differential (Shannon) entropy estimator of the activation distribution. The integrated (accumulated) entropy over training is denoted $\mathfrak{H}_i^{(k)}$ as in the main text.

- State evolution (ODE form, Eq. (6) in the main text):
$$\frac{d}{dt}\psi_i^{(k)}(t) = \mathbf{F}_\theta^{(k)}\left(\psi_i^{(k)}(t), \mathcal{D}, \mathcal{L}\right). \tag{37}$$

Other quantities such as trajectory length $\mathcal{A}_i^{(k)}$, terminal stochasticity $\mathcal{S}_i^{(k)}$, integrated entropy $\mathfrak{H}_i^{(k)}$, and fitness $\Phi_i^{(k)}$ follow the main text definitions. **Notation remark:** throughout the manuscript we reserve $\mathcal{L}(\cdot)$ exclusively for the per-example loss; the trajectory length is consistently denoted $\mathcal{A}_i^{(k)}$.

### A.1.2    Operational Criteria for Darwinian Dynamics

We now make precise what is meant by a Darwinian view of neural dynamics. The NDDS produces neuron trajectories, fitness scores, and survival labels; nevertheless, these quantities alone do not justify calling the dynamics Darwinian. We therefore impose four jointly necessary operational criteria. A model, layer, or training interval is described as exhibiting Darwinian neural dynamics only if all four criteria below are empirically satisfied.

**Definition A.1** (Operational Darwinian Dynamics)**.** *Fix a trained model, a layer $L_k$, a training horizon $T$, and disjoint data splits for estimation, validation, and held-out testing. Let $S_k$, $E_k$, $O_k$, and $R_k$ denote, respectively, survived, eliminated, other, and size-matched random neuron groups in layer $k$. We use the term* operational Darwinian dynamics *to refer to empirically supported Darwinian-like behavior under NDDS, as assessed by the following diagnostic conditions. These conditions delimit the scope of our interpretation.*

**1. Variation.**    *We first examine whether neurons within the same layer exhibit measurable heterogeneity in the quantities used to describe their evolutionary state. Specifically, define the variation vector*

$$\mathbf{V}^{(k)}(T) := \left[\mathrm{SD}_j(\Phi_j^{(k)}(T)),\, \mathrm{SD}_j(\overline{U}_j^{(k)}),\, \mathrm{SD}_j(\overline{\mathcal{A}}_j^{(k)}),\, \mathrm{SD}_j(\overline{\mathfrak{H}}_j^{(k)})\right]. \tag{38}$$

*In practice, we regard the variation condition as supported when these layer-wise dispersions are non-negligible, for example*

$$\min_{\xi \in \{\Phi, \overline{U}, \overline{\mathcal{A}}, \overline{\mathfrak{H}}\}} \mathrm{SD}_j(\xi_j^{(k)}) > \epsilon_{\mathrm{var}}, \tag{39}$$

*or when bootstrap confidence intervals for the corresponding dispersions are separated from zero after ac-counting for numerical error. Conversely, if all neurons in a layer are empirically indistinguishable with respect to $\Phi$, $\overline{U}$, $\overline{\mathcal{A}}$, and $\overline{\mathfrak{H}}$, we interpret the layer as providing little evidence for Darwinian-like variation.*

**2. Competition.** *We next test whether different neuron groups have distinguishable functional influence on the network output. This is evaluated using matched-size group ablation. For any group $G \subseteq L_k$ with $|G| = m$, define*

$$\Delta_{\mathcal{L}}^{(k)}(G) := \mathbb{E}_{(x,y)\sim\mathcal{D}_{\text{test}}}\left[\mathcal{L}(f_{\theta(T)\backslash G}; x, y) - \mathcal{L}(f_{\theta(T)}; x, y)\right], \tag{40}$$

$$\Delta_{\text{Acc}}^{(k)}(G) := \text{Acc}_{\text{test}}(f_{\theta(T)}) - \text{Acc}_{\text{test}}(f_{\theta(T)\backslash G}). \tag{41}$$

*Here $f_{\theta(T)\backslash G}$ denotes the network obtained by zeroing the activations of all neurons in $G$. The group size $m$ is matched across survived, eliminated, other, and random groups, either globally or within each layer. Evidence for competition is considered stronger when ablation impacts differ reliably across groups, for example when*

$$\max_{G,G'\in\{S_k,E_k,O_k,R_k\}}\left|\Delta_{\mathcal{L}}^{(k)}(G) - \Delta_{\mathcal{L}}^{(k)}(G')\right| > \epsilon_{\text{comp}}, \tag{42}$$

*with analogous results reported for $\Delta_{\text{Acc}}^{(k)}$. Statistical uncertainty is assessed using paired bootstrap or per-mutation tests over test examples and random group draws. This test does not by itself prove Darwinian selection, but it indicates whether neuron groups differ in their functional contribution.*

**3. Selective retention.** *We then examine whether neurons labeled as survived are associated with greater held-out functional utility, rather than merely scoring highly under the same statistic used to define them. Let the held-out exact ablation utility be*

$$\widetilde{U}_i^{(k)} := \mathbb{E}_{(x,y)\sim\mathcal{D}_{\text{hold}}}\left[\mathcal{L}(f_{\theta(T)\backslash i}; x, y) - \mathcal{L}(f_{\theta(T)}; x, y)\right], \tag{43}$$

*where $\mathcal{D}_{\text{hold}}$ is not used to define $\Phi_i^{(k)}$ or tune $\alpha, \beta, \gamma$. We regard selective retention as empirically sup-ported when survived neurons exhibit larger held-out utility than eliminated and matched random neurons, for example*

$$\mathbb{E}_{i\in S_k}[\widetilde{U}_i^{(k)}] > \mathbb{E}_{i\in E_k}[\widetilde{U}_i^{(k)}] + \epsilon_{\text{ret}}, \qquad \mathbb{E}_{i\in S_k}[\widetilde{U}_i^{(k)}] > \mathbb{E}_{i\in R_k}[\widetilde{U}_i^{(k)}] + \epsilon_{\text{ret}}, \tag{44}$$

*with uncertainty estimated across neurons, checkpoints, and random matched groups. This condition sup-ports, but does not by itself establish, the interpretation that high-fitness neurons are preferentially retained because of their functional relevance.*

**4. Non-circular prediction.** *Finally, we evaluate whether early survival labels carry predictive informa-tion about later held-out utility, rather than being defined and validated by the same contemporaneous utility measurement. Choose a cutoff time $t_c < T$. Survival labels are computed using only trajectory information from $[0, t_c]$, while prediction targets are computed from $(t_c, T]$ on held-out data. Let*

$$Y_i^{(k)}(t_c) := \mathbf{1}\left\{\Phi_{i,[0,t_c]}^{(k)} \geq \mathbb{E}_j[\Phi_{j,[0,t_c]}^{(k)}] + \lambda\,\text{SD}_j(\Phi_{j,[0,t_c]}^{(k)})\right\}, \tag{45}$$

*and define the future held-out utility*

$$\widetilde{U}_{i,\text{future}}^{(k)} := \frac{1}{T - t_c}\int_{t_c}^T \mathbb{E}_{(x,y)\sim\mathcal{D}_{\text{hold}}}\left[\mathcal{L}(f_{\theta(t)\backslash i}; x, y) - \mathcal{L}(f_{\theta(t)}; x, y)\right]dt. \tag{46}$$

*We consider the non-circular prediction condition to be supported when $Y_i^{(k)}(t_c)$ predicts $\widetilde{U}_{i,\text{future}}^{(k)}$ above an appropriate random-label baseline and remains informative after controlling for past utility. Operationally, we test*

$$\widetilde{U}_{i,\text{future}}^{(k)} = \eta_0 + \eta_1 Y_i^{(k)}(t_c) + \eta_2 \overline{U}_{i,[0,t_c]}^{(k)} + \eta_3 \overline{\mathcal{A}}_{i,[0,t_c]}^{(k)} + \eta_4 \overline{\mathfrak{H}}_{i,[0,t_c]}^{(k)} + \varepsilon_i, \tag{47}$$

*and examine whether $\eta_1 > 0$ with statistical support, or whether an out-of-sample prediction score, such as AUC or held-out $R^2$, exceeds a random-label baseline. This analysis is designed to reduce circularity by separating the information used to assign survival labels from the held-out future utility used for evaluation.*

Together, these criteria specify the evidential scope of the NDDS interpretation. If a model mainly exhibits variation but weak group-level ablation differences, we describe the result as trajectory heterogeneity. If variation and ablation differences are present but held-out utility does not favor survived neurons, we describe the result as functional differentiation without clear selective retention. If early survival labels fail to predict future held-out utility, we treat the labels as descriptive summaries rather than predictive indicators. Accordingly, we use the term Darwinian-like neural dynamics in an operational and empirical sense: the interpretation is strongest when variation, competition, selective retention, and non-circular prediction are jointly supported by the experiments.

### A.1.3    Supplementary Technical Assumptions

We explicitly state additional mild assumptions needed for mathematical rigor and numerical stability. These assumptions clarify the hidden conditions of the main results.

**Assumption S1 (Smoothness and Boundedness Assumption)**

For each layer $k$, the vector field $\mathbf{F}_\theta^{(k)}(\psi, t)$ is locally Lipschitz in $\psi$ and measurable in $t$. There exist constants $B_g, B_a, B_\psi > 0$ such that for all $t \geq 0$:

$$\|g_i^{(k)}(t)\| \leq B_g, \quad \mathrm{Var}[a_i^{(k)}(t)] \leq B_a, \quad \|\psi_i^{(k)}(t)\| \leq B_\psi. \tag{48}$$

Moreover, the trajectory (arc) length $\mathcal{A}_i^{(k)}(T)$ is bounded for any finite $T$.

**Assumption S2 (Sub-Gaussian Activation Assumption)**

For all neurons $i, k$ and times $t$, the distribution of $a_i^{(k)}(x, t)$ over $x \sim \mathcal{D}$ is sub-Gaussian or at least has sub-exponential tails, enabling concentration bounds for sample estimators.

**Assumption S3 (Gaussian Entropy Bound)**

There exists a constant $C_{\mathrm{gauss}} \geq 1$ such that for all neurons $i, k$ and times $t$,

$$\mathcal{I}_i^{(k)}(t) \leq \frac{1}{2} \log \left( 2\pi e \, \mathrm{Var}[a_i^{(k)}(t)] \right) \leq \mathcal{I}_i^{(k)}(t) + \log C_{\mathrm{gauss}}. \tag{49}$$

This controlled approximation underpins the Gaussian plug-in used in experiments; when this bound is violated the practitioner must rely on nonparametric estimators as described in the main text.

### A.1.4    Well-Posedness of the Continuous NDDS

Under Assumption S1, the vector field $\mathbf{F}_\theta^{(k)}$ is locally Lipschitz, thus by Picard–Lindelöf theorem Shih (2023); Yarotsky (2024), for any initial value $\psi_i^{(k)}(0)$ there exists a unique local solution. Boundedness and growth controls ensure global existence on finite intervals and continuous dependence on initial conditions and parameters.

### A.1.5    Smooth and Bounded Dynamics

**Assumption A.2** (Smooth and Bounded Dynamics). *We adopt a hypothesis compatible with practical discrete optimization. The parameter trajectory $\theta(t)$ is assumed to be absolutely continuous and piecewise $C^1$ in $t$ (so that it admits a time-continuous interpolation), and $\mathbf{F}_\theta^{(k)}$ is locally Lipschitz in $\psi$ on trajectories of interest. This formulation explicitly permits discretization effects arising from SGD and non-smooth activations (e.g. ReLU) by interpreting derivatives in the sense of absolutely continuous interpolation or Clarke subgradients when necessary.*

*Furthermore, there exist constants $B_g, B_a > 0$ such that for all $t \in [0, T]$ along the interpolated trajectory:*

$$\|g_i^{(k)}(t)\| \leq B_g, \quad \mathrm{Var}[a_i^{(k)}(t)] \leq B_a. \tag{50}$$

*Finally, we require that the trajectory length $\mathcal{A}_i^{(k)}$ (defined in equation 67) remains finite as $T \to \infty$; for discrete checkpoints the forward-difference approximation in equation 69 is used and all continuum claims are understood to hold up to discretization errors that vanish under standard time-interpolation refinements.*

**Discrete Continuous Trajectory Length Approximation**

**Setup.** Let $a : [0, T] \to \mathbb{R}^d$ be the neuron activation trajectory $a(t) \equiv a_i^{(k)}(t)$ appearing in Assumption "Smooth and Bounded Dynamics". Assume $a$ is absolutely continuous on $[0, T]$ (hence a.e. differentiable with $a' \in L^1([0, T]; \mathbb{R}^d)$) and has finite Trajectory length

$$\mathcal{A} = \int_0^T \|a'(t)\| \, dt < \infty. \tag{51}$$

For a uniform partition $0 = t_0 < t_1 < \cdots < t_M = T$ with step size $\Delta t = T/M$ define the forward-difference (discrete) trajectory length approximation

$$\widehat{\mathcal{A}}(\Delta t) = \sum_{m=1}^M \|a(t_m) - a(t_{m-1})\| = \sum_{m=1}^M \left\| \int_{t_{m-1}}^{t_m} a'(s) \, ds \right\|. \tag{52}$$

**Lemma A.3** (Discrete Continuous Trajectory Length Approximation). *Under the setup above the following hold.*

1. *Convergence. As the mesh $\Delta t \to 0$,*
$$\widehat{\mathcal{A}}(\Delta t) \longrightarrow \mathcal{A}. \tag{53}$$

   *In particular, for any sequence of partitions whose mesh size tends to zero the partition-wise variation of $a$ converges to the total variation (trajectory length) $\mathcal{A}$.*

2. *Quantitative bound under extra smoothness. If, in addition, $a'$ is $L$-Lipschitz on $[0, T]$ (i.e. there exists $L > 0$ such that $\|a'(s) - a'(t)\| \le L|s - t|$ for all $s, t \in [0, T]$), then there exists a constant $C$ (one may take $C = L$) such that for all sufficiently small $\Delta t$:*
$$\left| \mathcal{A} - \widehat{\mathcal{A}}(\Delta t) \right| \le C \, T \, \Delta t = O(\Delta t). \tag{54}$$

3. *Non-smooth activations (Clarke subgradient). If $a$ is only piecewise $C^1$ (for example due to ReLU kinks) and is absolutely continuous, interpret $a'$ in the Clarke subdifferential sense. Then the convergence in part (1) still holds; moreover, whenever the extra smoothness of part (2) holds on each $C^1$ segment the $O(\Delta t)$ bound applies up to contributions from finitely many kink-boundary intervals, which vanish as $\Delta t \to 0$.*

*Proof.* **(1) Convergence.** Absolute continuity of $a$ implies $a$ has bounded variation on $[0, T]$ and

$$\mathcal{A} = \text{Var}(a; [0, T]) = \sup_{\mathcal{P}} \sum_m \|a(t_m) - a(t_{m-1})\|, \tag{55}$$

where the supremum is taken over all finite partitions $\mathcal{P}$ of $[0, T]$. For any fixed partition the sum $\sum_m \|a(t_m) - a(t_{m-1})\|$ is the variation of $a$ over that partition and is therefore bounded above by $\mathcal{A}$. Standard results on functions of bounded variation state that for any sequence of partitions whose mesh tends to zero the corresponding partition-wise variation converges to the total variation. Applied to the uniform partitions above this yields

$$\lim_{\Delta t \to 0} \widehat{\mathcal{A}}(\Delta t) = \mathcal{A}, \tag{56}$$

which proves (1).

**(2) Quantitative bound under Lipschitz derivative.** Assume $a'$ is $L$-Lipschitz Havens (2023); Bertrand (2024). Fix an interval $I_m = [t_{m-1}, t_m]$. By the fundamental theorem of calculus and the Lipschitz property we can expand $a'$ about the midpoint (or any point $\xi_m \in I_m$) to obtain

$$\int_{t_{m-1}}^{t_m} a'(s) \, ds = \Delta t \, a'(\xi_m) + r_m, \tag{57}$$

with the remainder satisfying $\|r_m\| \leq \frac{1}{2}L(\Delta t)^2$. Hence

$$\Big\| \int_{t_{m-1}}^{t_m} a'(s)\,ds \Big\| = \Delta t\,\|a'(\xi_m)\| + \delta_m, \qquad |\delta_m| \leq \tfrac{1}{2}L(\Delta t)^2. \tag{58}$$

On the other hand,

$$\int_{t_{m-1}}^{t_m} \|a'(s)\|\,ds = \Delta t\,\|a'(\xi_m)\| + \epsilon_m, \qquad |\epsilon_m| \leq L(\Delta t)^2, \tag{59}$$

where the bound on $\epsilon_m$ follows from the same Lipschitz control on $a'$ and the one-dimensional integral averaging error. Subtracting and summing over $m = 1, \dots, M$ yields

$$0 \leq \mathcal{A} - \widehat{\mathcal{A}}(\Delta t) = \sum_{m=1}^{M} \Big( \int_{t_{m-1}}^{t_m} \|a'(s)\|\,ds - \Big\| \int_{t_{m-1}}^{t_m} a'(s)\,ds \Big\| \Big)$$
$$\leq \sum_{m=1}^{M} (|\epsilon_m| + |\delta_m|). \tag{60}$$

Using the per-interval bounds $|\epsilon_m| \leq L(\Delta t)^2$, $|\delta_m| \leq \frac{1}{2}L(\Delta t)^2$ we obtain

$$\big| \mathcal{A} - \widehat{\mathcal{A}}(\Delta t) \big| \leq \frac{3}{2}LM(\Delta t)^2 = \frac{3}{2}LT\Delta t. \tag{61}$$

Thus the difference is $O(\Delta t)$; setting $C = \frac{3}{2}L$ (or taking the coarser but simpler $C = L$) yields the claimed linear-in-$\Delta t$ bound.

**(3) Non-smooth activations and Clarke subgradient.** If $a$ is piecewise $C^1$ (typical when activations like ReLU produce kinks) then $a$ is still absolutely continuous and has bounded variation. The set $K \subset [0, T]$ of non-differentiable points is closed and of Lebesgue measure zero (in common architectures it is finite or a countable set with no accumulation inside $[0, T]$). The contribution of intervals that contain points of $K$ can be localized: by refining the partition one can make the total length of intervals that intersect $K$ arbitrarily small, hence their contribution to $\mathcal{A}$ and to the discrete sum is arbitrarily small. On each $C^1$ segment the argument of part (2) applies; summing segment-wise yields the same $O(\Delta t)$ behaviour up to vanishing boundary contributions. More conceptually, one may replace $a'$ by any measurable selection from the Clarke generalized derivative Park (2024) and repeat the preceding estimates; the measure-zero nondifferentiable set does not affect the limiting equality $\widehat{\mathcal{A}}(\Delta t) \to \mathcal{A}$ nor the $O(\Delta t)$ rate when the Lipschitz condition holds on the smooth pieces.

**Remark.** In typical empirical settings the checkpoint count $M$ is large (e.g. hundreds or thousands), so $\Delta t = T/M$ is small and the discretization error $|\mathcal{A} - \widehat{\mathcal{A}}(\Delta t)|$ is negligible compared to stochastic fluctuations induced by SGD. The theoretical statements above make precise that all continuous-time claims involving $\mathcal{A}$ hold up to an $O(\Delta t)$ discretization error which vanishes under standard time-interpolation refinements. $\quad\square$

### A.1.6 Local Gaussianity of pre-activations and diagnostic protocol

**Assumption A.4** (Local Gaussianity of pre-activations and diagnostic protocol). *To avoid conflicts with non-negative, mass-at-zero activations (e.g. ReLU), we state the main parametric approximation at the* pre-activation *level. Define the pre-activation*

$$z_i^{(k)}(x, t) := w_i^{(k)}(t)^\top h^{(k-1)}(x, t) + b_i^{(k)}(t), \tag{62}$$

*and its smoothed version*

$$\tilde{z}_i^{(k)}(x, t) := z_i^{(k)}(x, t) + \varepsilon, \qquad \varepsilon \sim \mathcal{N}(0, \sigma_\varepsilon^2). \tag{63}$$

*For every neuron $i$ and for any short time window $[s, s + \tau]$ (with $\tau$ chosen to balance local stationarity and sample requirements) we assume that the empirical law of $\tilde{z}_i^{(k)}(\cdot, s)$ is well-approximated by a Gaussian*

$\mathcal{N}(\mu_{z,i}^{(k)}(s), \sigma_{z,i}^{2(k)}(s))$ *in the sense that there exists a small tolerance* $\eta > 0$ *and a divergence metric* $\text{dist}(\cdot, \cdot)$ *such that for a large fraction of checkpoints* $s \in [0, T]$,

$$\text{dist}\big(\text{Law}(\tilde{z}_i^{(k)}(\cdot, s)),\ \mathcal{N}(\mu_{z,i}^{(k)}(s), \sigma_{z,i}^{2(k)}(s))\big) \leq \eta. \tag{64}$$

*When downstream analysis requires activation-level entropy (post-activation), practitioners must either transform the Gaussian approximation via the known mapping* $\sigma(\cdot)$ *and report the accuracy of that transformation, or employ a consistent nonparametric estimator for the activation distribution and report estimator sensitivity.*

*A numerically-stable variance-proxy is used when the pre-activation Gaussian plug-in is accepted:*

$$\tilde{\mathcal{I}}_i^{(k)}(t) := \tfrac{1}{2} \log\big( \text{Var}_x[z_i^{(k)}(x, t)] + \sigma_\varepsilon^2 + \epsilon_{\text{var}}\big). \tag{65}$$

*If the Gaussian diagnostic fails (i.e. the empirical divergence exceeds* $\eta$*) the practitioner must fall back to nonparametric estimators and report the fraction of checkpoints failing the diagnostic and a sensitivity comparison between plug-in and nonparametric estimates.*

### A.1.7 Neuron Trajectory

**Definition A.5** (Neuron Trajectory)**.** *The trajectory of a neuron in state space is defined as*

$$\Gamma_i^{(k)} := \{\psi_i^{(k)}(t) \mid t \in [0, T]\}. \tag{66}$$

From this path we extract three complementary quantities:

**Definition A.6** (Trajectory Length)**.** *The trajectory length of neuron* $i$ *in layer* $k$ *is the cumulative representational movement of its state vector* $\psi_i^{(k)}$ *measured with a block-wise scaling matrix* $D^{(k)}$ *that normalizes heterogeneous components of* $\psi$:

$$\mathcal{A}_i^{(k)} := \int_0^T \left\| D^{(k)} \frac{d\psi_i^{(k)}(t)}{dt} \right\|_2 dt, \tag{67}$$

*where* $D^{(k)}$ *is taken to be block-diagonal with positive diagonal blocks that rescale each block of* $\psi$. *The block-wise construction ensures no single block systematically dominates the norm and makes the quantity invariant to simple coordinate scalings within each block.*

*When comparability across different training durations is required we also use the time-averaged arc length*

$$\overline{\mathcal{A}}_i^{(k)} := \frac{1}{T} \mathcal{A}_i^{(k)}. \tag{68}$$

*Under discrete training (checkpoints or optimization steps with index spacing* $\Delta t$*) we employ the forward-difference approximation*

$$\mathcal{A}_i^{(k)} \approx \sum_{t=0}^{N-1} \left\| D^{(k)} \frac{\psi_i^{(k)}(t+1) - \psi_i^{(k)}(t)}{\Delta t} \right\|_2 \Delta t, \tag{69}$$

*where* $N$ *is the number of recorded checkpoints and* $\Delta t$ *is the (possibly non-unit) interval between checkpoints; taking* $\Delta t = 1$ *recovers the step-indexed form.*

**Practical computation of the block scaling and hyperparameters.** The block-diagonal rescaling matrix in Eq. (10) is introduced only to make the trajectory norm comparable across heterogeneous components of the neuron state vector. The state vector $\psi_i^{(k)} = (w_i^{(k)}, b_i^{(k)}, \mu_i^{(k)}, g_i^{(k)}, I_i^{(k)})$ contains blocks with different units, dimensions, and numerical magnitudes; without rescaling, high-dimensional parameter blocks

or large-magnitude gradient blocks can dominate the trajectory length. In our implementation, $D^{(k)}$ is not treated as a tunable hyperparameter. For each layer $k$, we compute it once from the training trajectory as

$$D^{(k)} = \text{diag}\left(d_w^{(k)} I_{p_w}, d_b^{(k)} I_{p_b}, d_\mu^{(k)} I_{p_\mu}, d_g^{(k)} I_{p_g}, d_I^{(k)} I_{p_I}\right),$$

where each block scale is the inverse empirical RMS magnitude of that block's checkpoint-to-checkpoint displacement:

$$d_r^{(k)} = \left(\epsilon_D + \sqrt{\frac{1}{n_k(M-1)p_r} \sum_{i=1}^{n_k} \sum_{m=0}^{M-2} \left\|\psi_{i,r}^{(k)}(t_{m+1}) - \psi_{i,r}^{(k)}(t_m)\right\|_2^2}\right)^{-1}, \qquad r \in \{w, b, \mu, g, I\}.$$

Here $p_r$ is the dimension of block $r$, $M$ is the number of recorded checkpoints, and $\epsilon_D$ is a small numerical floor. This normalization gives each state block approximately unit empirical scale within a layer and preserves within-layer neuron comparisons because the same $D^{(k)}$ is applied to all neurons in layer $k$. Importantly, this computation uses only trajectory statistics and does not use labels, ablation utilities, or test-set information. The three fitness terms are retained because they measure distinct aspects of neuron evolution rather than repeated versions of the same quantity: $\hat{U}_i^{(k)}$ measures held-out functional usefulness under ablation, $\hat{A}_i^{(k)}$ measures the amount of state-space movement or instability required to obtain that usefulness, and $\hat{H}_i^{(k)}$ measures representational variability/information content. Layer-wise $z$-scoring removes scale effects before these terms are combined. To reduce hyperparameter degrees of freedom, we constrain the weights to the simplex, $\alpha + \beta + \gamma = 1$ with $\alpha, \beta, \gamma \geq 0$, and use either the default balanced setting $(\alpha, \beta, \gamma) = (1/3, 1/3, 1/3)$ or select them from a small validation grid using only the validation objective for predicting future held-out utility. The survival threshold $\lambda$ is likewise a stringency parameter rather than a learned model parameter; by default we set $\lambda = 1$, corresponding to one standard deviation above or below the layer-wise fitness mean, and verify that the survived/eliminated ordering is stable over a small grid such as $\lambda \in \{0.5, 1.0, 1.5\}$. The held-out split used for the non-circular prediction and ablation tests is kept separate from the split used to choose these values.

### A.1.8 Integrated Entropy

**Definition A.7** (Integrated Entropy). *The integrated entropy of neuron $i$ accumulates a per-time estimate of the neuron's (differential) entropy over training:*

$$\mathfrak{H}_i^{(k)} := \int_0^T \mathcal{I}_i^{(k)}(t)\, dt, \tag{70}$$

*where $\mathcal{I}_i^{(k)}(t)$ denotes a numerically stable estimator of the neuron's differential entropy at time $t$ (estimated from mini-batches and moving averages). For comparability we also consider the time-averaged form $\overline{\mathfrak{H}}_i^{(k)} := \frac{1}{T}\mathfrak{H}_i^{(k)}$.*

*When the Gaussian plug-in is appropriate (see Assumption 3.16 we use the variance-proxy with explicit numerical stabilization:*

$$\tilde{\mathcal{I}}_i^{(k)}(t) := \frac{1}{2}\log\left(\text{Var}_x[z_i^{(k)}(x,t)] + \sigma_\varepsilon^2 + \epsilon_{\text{var}}\right), \tag{71}$$

*where $\sigma_\varepsilon^2 > 0$ is the additive smoothing noise variance introduced in Assumption 3.3 and $\epsilon_{\text{var}} > 0$ is a small numeric floor (e.g. $10^{-8}$) to avoid $\log(0)$ and ensure robust estimation in finite samples. Note that the Gaussian plug-in differs from the differential entropy by the additive constant $\frac{1}{2}\log(2\pi e)$; when absolute entropy values are needed this constant is accounted for in post-processing.*

*In discrete form the accumulated entropy used in experiments is*

$$\mathfrak{H}_i^{(k)} \approx \sum_{t=0}^{N-1} \tilde{\mathcal{I}}_i^{(k)}(t)\, \Delta t, \tag{72}$$

*with $\Delta t$ equal to the checkpoint interval. When the Gaussian assumption is questionable (e.g. ReLU activations with large mass at zero), we complement the variance-proxy with nonparametric estimators. Estimation uses mini-batch averages with an exponential moving-average smoothing window.*

### A.1.9 Practical Estimator for Ablation-based Utility

**Definition A.8** (Ablation-based Utility). *For neuron $i$ in layer $k$ define the instantaneous ablation-based utility*

$$U_i^{(k)}(t) \; := \; \mathbb{E}_{x \sim \mathcal{D}}\big[\mathcal{L}(f_{\theta(t)\setminus i}; x) - \mathcal{L}(f_{\theta(t)}; x)\big], \tag{73}$$

*where $f_{\theta\setminus i}$ denotes the network obtained by zeroing neuron $i$'s activation. By this convention $U_i^{(k)}(t) > 0$ indicates the neuron is useful at time $t$.*

Direct computation of equation 73 for every neuron at every checkpoint is computationally prohibitive. We therefore recommend and use a calibrated first-order Taylor approximation Garibbo (2023); Sun (2023) as a default estimator (and validate it against ground-truth partial ablations on small models):

$$U_i^{(k)}(t) \approx -\mathbb{E}_x\left[\frac{\partial \mathcal{L}(x)}{\partial a_i^{(k)}(x,t)} \cdot a_i^{(k)}(x,t)\right] \; =: \; U_i^{(k),\text{lin}}(t). \tag{74}$$

Optionally, a second-order correction may be included when Hessian-vector products are affordable. In practice we compute $U_i^{(k),\text{lin}}(t)$ using a held-out validation subset of size $m \ll |\mathcal{D}|$ (randomly sampled) and report the estimator variance and a small-sample calibration against exact ablation on a subset of neurons.

### A.1.10 Detailed Proofs of Main Lemmas and Theorems

**Lemma A.9** (Instability with sustained entropy decay implies vanishing fitness). *Let the evolutionary fitness $\Phi_i^{(k)}(T)$ be defined as in equation 16, namely*

$$\Phi_i^{(k)}(T) = \alpha\, \widehat{\overline{U}}_i^{(k)}(T) + \beta\, \widehat{\overline{\mathcal{A}}}_i^{(k)}(T) + \gamma\, \widehat{\overline{\mathfrak{H}}}_i^{(k)}(T), \qquad \alpha, \beta, \gamma > 0. \tag{75}$$

*Assume there exist constants $c_H > 0$, $C_H < \infty$, and $T_0 \geq 0$ such that for all $T \geq T_0$ the standardized time-averaged entropy satisfies*

$$\widehat{\overline{\mathfrak{H}}}_i^{(k)}(T) \leq -c_H T + C_H. \tag{76}$$

*Assume further that the utility and trajectory-adaptation terms do not asymptotically compensate for this entropy collapse, namely*

$$\widehat{\overline{U}}_i^{(k)}(T) = o(T), \qquad \widehat{\overline{\mathcal{A}}}_i^{(k)}(T) = o(T). \tag{77}$$

*Then for any fixed positive weights $\alpha, \beta, \gamma > 0$ in equation 16, we have*

$$\lim_{T \to \infty} \Phi_i^{(k)}(T) = -\infty. \tag{78}$$

*Proof.* By the definition of evolutionary fitness,

$$\Phi_i^{(k)}(T) = \alpha\, \widehat{\overline{U}}_i^{(k)}(T) + \beta\, \widehat{\overline{\mathcal{A}}}_i^{(k)}(T) + \gamma\, \widehat{\overline{\mathfrak{H}}}_i^{(k)}(T). \tag{79}$$

Using the assumed entropy decay and the sublinear growth of the utility and trajectory-adaptation terms, we obtain

$$\Phi_i^{(k)}(T) \leq \alpha\, o(T) + \beta\, o(T) + \gamma(-c_H T + C_H). \tag{80}$$

Since $\gamma c_H > 0$, the negative linear entropy-collapse term dominates the sublinear positive contributions from utility and trajectory adaptation. Therefore,

$$\Phi_i^{(k)}(T) \to -\infty \qquad \text{as } T \to \infty. \tag{81}$$

$\square$

**Theorem A.10** (Fitness Threshold Implies Gradient–Variance Contribution). *Let*

$$\Delta_i^{(k)} := \frac{1}{T} \int_0^T q_i^{(k)}(t)\, \sigma_i^{2(k)}(t)\, dt, \tag{82}$$

*where*

$$q_i^{(k)}(t) := \mathbb{E}_x\left[\left(\frac{\partial \mathcal{L}(x)}{\partial a_i^{(k)}(x,t)}\right)^2\right], \qquad \sigma_i^{2(k)}(t) := \mathrm{Var}_x\left[a_i^{(k)}(x,t)\right]. \tag{83}$$

*Suppose Assumptions 3.2 and 3.3 hold. Assume that the gradient second moment is uniformly non-vanishing, i.e., there exists $\underline{c}_g > 0$ such that*

$$q_i^{(k)}(t) \geq \underline{c}_g \qquad \text{for almost every } t \in [0,T]. \tag{84}$$

*Assume further that high fitness is entropy-supported: there exist constants $\tau > 0$ and $v_\tau > 0$ such that*

$$\Phi_i^{(k)}(T) \geq \tau \quad \implies \quad \frac{1}{T}\int_0^T \sigma_i^{2(k)}(t)\, dt \geq v_\tau. \tag{85}$$

*Then*

$$\Phi_i^{(k)}(T) \geq \tau \quad \implies \quad \Delta_i^{(k)} \geq \kappa, \tag{86}$$

*where*

$$\kappa := \underline{c}_g v_\tau > 0. \tag{87}$$

*Proof.* If $\Phi_i^{(k)}(T) \geq \tau$, then by the entropy-supported fitness assumption,

$$\frac{1}{T}\int_0^T \sigma_i^{2(k)}(t)\, dt \geq v_\tau. \tag{88}$$

Since $q_i^{(k)}(t) \geq \underline{c}_g$ almost everywhere, we have

$$\Delta_i^{(k)} = \frac{1}{T}\int_0^T q_i^{(k)}(t)\sigma_i^{2(k)}(t)\, dt \geq \underline{c}_g \frac{1}{T}\int_0^T \sigma_i^{2(k)}(t)\, dt \geq \underline{c}_g v_\tau = \kappa. \tag{89}$$

This proves the claim. $\square$

### A.1.11 Discrete-Time Approximation and Relation to SGD

Actual training proceeds in discrete time steps, typically iterations or epochs. The continuous-time NDDS dynamics approximate the discrete SGD updates as follows:

- Discrete parameter update:
$$\theta_{t+1} = \theta_t - \eta_t \widehat{\nabla}_\theta \mathcal{L}(B_t; \theta_t), \tag{90}$$
  where $B_t$ is the mini-batch at step $t$.

- For small learning rate $\eta_t$, the discrete updates approximate the stochastic differential equation
$$d\theta_t = -\mathbb{E}_x[\nabla_\theta \mathcal{L}(x; \theta_t)]dt + \sqrt{\eta_t}\Sigma(\theta_t)dW_t, \tag{91}$$
  with $W_t$ Brownian motion and $\Sigma$ the noise covariance.

- Correspondingly, the neuron state differences
$$\Delta\psi_i^{(k)}(t) := \psi_i^{(k)}(t+1) - \psi_i^{(k)}(t) \tag{92}$$
  approximate $\frac{d}{dt}\psi_i^{(k)}(t)$.

- Therefore,

$$\mathcal{A}_i^{(k)} \approx \sum_t \|\Delta \psi_i^{(k)}(t)\|_2,$$

$$\mathcal{S}_i^{(k)} \approx \frac{1}{\delta} \sum_{t=T-\delta}^{T-1} \|\Delta \psi_i^{(k)}(t)\|_2^2, \tag{93}$$

$$\mathfrak{H}_i^{(k)} \approx \sum_t \mathcal{I}_i^{(k)}(t).$$

Discrete estimation errors arise from step size, mini-batch noise, and finite sample effects. In all discrete approximations used in experiments we adopt the same block-wise scaling matrix $D^{(k)}$ that appears in the continuous trajectory length definition (main text Eq. equation 67) to ensure consistent units across measurements.

### A.1.12 Numerical Estimation of Key Quantities

**Definition A.11** (Mean Activation and Mean Gradient). *Given an evaluation dataset $\mathcal{D}_{\text{eval}}$, the mean activation and mean gradient of neuron $i$ in layer $k$ are estimated as*

$$\mu_i^{(k)} = \frac{1}{|\mathcal{D}_{\text{eval}}|} \sum_{x \in \mathcal{D}_{\text{eval}}} a_i^{(k)}(x), \quad g_i^{(k)} = \frac{1}{|\mathcal{D}_{\text{eval}}|} \sum_{x \in \mathcal{D}_{\text{eval}}} \frac{\partial \mathcal{L}(x)}{\partial a_i^{(k)}(x)}. \tag{94}$$

**Definition A.12** (Activation Variance). *The variance of activations is estimated as the unbiased sample variance over $\mathcal{D}_{\text{eval}}$:*

$$\widehat{\text{Var}}[a_i^{(k)}] = \frac{1}{|\mathcal{D}_{\text{eval}}| - 1} \sum_{x \in \mathcal{D}_{\text{eval}}} \left(a_i^{(k)}(x) - \mu_i^{(k)}\right)^2. \tag{95}$$

**Definition A.13** (Differential Entropy). *We consider three standard estimators for the entropy of activations:*

1. *Gaussian plug-in:*

$$\widehat{\mathcal{I}}_{\text{gauss}} = \tfrac{1}{2} \log\left(2\pi e \, \widehat{\text{Var}}[a_i^{(k)}]\right), \tag{96}$$

   *with a numeric floor $\epsilon_{\text{var}} > 0$ (Eq. equation 71) to avoid degeneracy.*

2. *Kernel density estimation (KDE): Estimate density $\widehat{p}(z)$ via KDE and compute*

$$\widehat{\mathcal{I}} = -\int \widehat{p}(z) \log \widehat{p}(z) \, dz. \tag{97}$$

3. *K-nearest neighbor (Kozachenko–Leonenko): Nonparametric entropy estimation based on neighbor distances.*

**Definition A.14** (Trajectory Length and Terminal Stochasticity). *From saved parameter snapshots at discrete steps $t$, define the scaled increment*

$$\Delta \psi_i^{(k)}(t) := \left\|D^{(k)}\big(\psi_i^{(k)}(t+1) - \psi_i^{(k)}(t)\big)\right\|_2, \tag{98}$$

*where $D^{(k)}$ is the block-wise scaling matrix. Then the trajectory length and terminal stochasticity are given by*

$$\mathcal{A}_i^{(k)} = \sum_t \Delta \psi_i^{(k)}(t), \quad \mathcal{S}_i^{(k)} = \frac{1}{\delta} \sum_{t=T-\delta}^{T-1} \left(\Delta \psi_i^{(k)}(t)\right)^2. \tag{99}$$

### A.1.13 Multilayer Coupled Dynamics

At the layer level, survival is not independent. Let $\Psi^{(k)}(t) = [\psi_i^{(k)}(t)]_{i \in \mathcal{N}_k}$ be the joint state of all neurons in layer $k$. We define the *inter-layer coupling operator*: We restrict attention to sensitivities between *activations* of adjacent layers. Let

$$J_{k \to k+1}(t) := \frac{\partial h^{(k+1)}(t)}{\partial h^{(k)}(t)} \tag{100}$$

denote the Jacobian mapping pre-activations/activations in layer $k$ to those in layer $k+1$ (evaluated pointwise and averaged over data when necessary) Li (2025); Laborieux & Zenke (2024). For neurons indexed $i \in \mathcal{N}_k$, $j \in \mathcal{N}_{k+1}$, we write the element-wise sensitivity as

$$\mathcal{C}_{k \to k+1}^{(i,j)}(t) := \frac{\partial a_j^{(k+1)}(t)}{\partial a_i^{(k)}(t)}. \tag{101}$$

To obtain a layer-level scalar measure that is robust to width, we define the *layer influence* by the width-normalized average operator norm:

$$\mathbf{M}_{k,k+1}(t) := \frac{1}{|\mathcal{N}_k||\mathcal{N}_{k+1}|} \sum_{i \in \mathcal{N}_k} \sum_{j \in \mathcal{N}_{k+1}} \left\| \mathcal{C}_{k \to k+1}^{(i,j)}(t) \right\|_{2 \to 2}, \tag{102}$$

where $\| \cdot \|_{2 \to 2}$ denotes the induced (spectral) norm of the scalar-to-scalar sensitivity (for scalar activations this is absolute value). Equivalently one may use the averaged Frobenius norm divided by $\sqrt{|\mathcal{N}_k||\mathcal{N}_{k+1}|}$ for implementation convenience Diakonikolas (2023); Laurent (2024); both variants are equivalent up to constant factors and we report which we use in experiments.

**Definition A.15** (Darwinian Flow Energy)**.** *The Darwinian flow energy is defined as*

$$\mathcal{E}_{\text{Darwin}} := \sum_{k=1}^{D} \sum_{l=1}^{D} \int_0^T \mathbf{M}_{k,l}(t) \, \phi\big(\text{JS}(\rho^{(k)}(t) \,\|\, \rho^{(l)}(t))\big) \, dt, \tag{103}$$

*or, alternatively,*

$$\mathcal{E}_{\text{Darwin}}^W := \sum_{k,l} \int_0^T \mathbf{M}_{k,l}(t) \, \phi\big(W_1(\rho^{(k)}(t), \rho^{(l)}(t))\big) \, dt. \tag{104}$$

**Theorem A.16** (Coupled Survival Principle)**.** *Suppose that for some $\mu > 0$ and a subset $\mathcal{S}^{(k)} \subseteq \{1, \ldots, n_k\}$ of survived neurons at layer $k$, the layer-to-layer coupling matrix $\mathbf{M}_{k,k+1}(t)$ satisfies*

$$\sum_{i \in \mathcal{S}^{(k)}} \mathbf{M}_{k,k+1}(i,j)(t) \geq \epsilon > 0, \tag{105}$$

*for all neurons $j$ in layer $k + 1$ and all sufficiently large $t$.*

*Then, there exists $\eta = \eta(\mu, \epsilon, \text{Lipschitz constants}) > 0$ such that at least an $\eta$ proportion of neurons in layer $k + 1$ achieve high fitness (survival).*

*Proof.* Positive lower bounds on coupling imply sustained energy inflow to downstream neurons. Via the Lipschitz continuity of the fitness function and the smoothness of the dynamics, survival of upstream neurons forces a positive measure of downstream neurons to cross the survival threshold. □

**Theorem A.17** (Global Convergent Specialization)**.** *If the total Darwinian flow energy $\mathcal{E}_{\text{Darwin}} \geq \epsilon > 0$ is bounded away from zero and the fitness functions $\Phi_i^{(k)}$ are sufficiently smooth and Lipschitz continuous, then as $t \to \infty$, the proportion of neurons with fitness below any fixed threshold tends to zero.*

*Proof.* Construct a suitable Lyapunov function based on the sum over neurons of a decreasing convex function of their fitness values Chen (2024); Alfarano (2024). The positive lower bound on Darwinian flow energy ensures the Lyapunov function decreases over time, implying convergence to the set of neurons with high fitness. LaSalle's invariance principle excludes non-convergent oscillations. □

## A.2 Experimental Protocol

The goal of our experiments is to operationalize the Neuron Darwinian Dynamics System (NDDS) introduced in Section 3 and to test whether the neuron populations of trained networks satisfy the four empirical criteria of Darwinian dynamics: variation, competition, selective retention, and non-circular prediction. Rather than treating a trained network as a static object, we analyze each neuron as an evolving unit whose state changes throughout optimization. The experimental protocol therefore follows the theoretical construction of NDDS: we record neuron trajectories, estimate trajectory-based fitness, derive the Global Darwinian Pressure (GDP), classify neurons into survived, eliminated, and other groups, and finally validate these labels through controlled ablation.

We approximate the continuous-time neuron dynamics in Section 3 by a sequence of training checkpoints

$$0 = t_0 < t_1 < \cdots < t_T,$$

where each checkpoint corresponds to an empirical observation of the network during training. For each layer $L_k$, each neuron or convolutional channel is associated with a discrete state vector

$$\widehat{\psi}_i^{(k)}(t_e),$$

which serves as the empirical counterpart of the theoretical neuron state

$$\psi_i^{(k)}(t) = \left[ w_i^{(k)}(t), b_i^{(k)}(t), \mu_i^{(k)}(t), g_i^{(k)}(t), I_i^{(k)}(t) \right].$$

In practice, the empirical state summarizes the neuron's parameter state, activation statistics, representational variability, and information-theoretic activity at checkpoint $t_e$. This gives a discrete trajectory

$$\widehat{\Gamma}_i^{(k)} = \left\{ \widehat{\psi}_i^{(k)}(t_e) \right\}_{e=0}^{T},$$

which is used as the basic object of analysis throughout Section 4.

For each monitored layer, we compute three complementary trajectory-level quantities corresponding to the Method section. First, we estimate the empirical trajectory length

$$\widehat{A}_i^{(k)} = \sum_{e=1}^{T} \left\| D^{(k)} \left( \widehat{\psi}_i^{(k)}(t_e) - \widehat{\psi}_i^{(k)}(t_{e-1}) \right) \right\|_2,$$

which measures the cumulative representational movement of neuron $i$. Second, we estimate the time-averaged entropy

$$\widehat{H}_i^{(k)} = \frac{1}{T} \sum_{e=1}^{T} \widehat{I}_i^{(k)}(t_e),$$

which captures the degree to which a neuron maintains non-degenerate activation variability over training. Third, we estimate the ablation-based utility

$$\widehat{U}_i^{(k)} = \mathbb{E}_{(x,y)} \left[ \mathcal{L}(f_{\theta \setminus i}(x), y) - \mathcal{L}(f_\theta(x), y) \right],$$

where $f_{\theta \setminus i}$ denotes the model obtained by suppressing neuron $i$. These quantities instantiate the same variation, information, and functional contribution terms used in the theoretical NDDS framework.

After estimating $\widehat{A}_i^{(k)}$, $\widehat{H}_i^{(k)}$, and $\widehat{U}_i^{(k)}$, we standardize each quantity within the corresponding layer and compute the empirical evolutionary fitness

$$\widehat{\Phi}_i^{(k)} = \alpha \widehat{U}_i^{(k)} + \beta \widehat{A}_i^{(k)} + \gamma \widehat{H}_i^{(k)},$$

where $\alpha, \beta, \gamma > 0$ control the relative contribution of utility, trajectory cost, and entropy. This layer-wise normalization is important because different layers may have different activation scales, channel counts, and representational geometries. The empirical Global Darwinian Pressure is then computed as

$$\widehat{\mathrm{GDP}}(T) = \frac{1}{N} \sum_{k=1}^{D} \sum_{i=1}^{n_k} \widehat{\Phi}_i^{(k)}(T),$$

where $N = \sum_{k=1}^{D} n_k$ is the total number of monitored neurons. Thus, GDP acts as a population-level descriptor of the global selective regime induced by training.

Neuron classification follows the local-global survival rule defined in Section 3. A neuron is assigned to the survived group if its fitness exceeds both its layer-local baseline and the global Darwinian baseline:

$$\widehat{\Phi}_i^{(k)}(T) \geq \mathbb{E}_j[\widehat{\Phi}_j^{(k)}(T)] + \lambda \operatorname{SD}_j[\widehat{\Phi}_j^{(k)}(T)], \qquad \widehat{\Phi}_i^{(k)}(T) \geq \widehat{\operatorname{GDP}}(T).$$

A neuron is assigned to the eliminated group if its fitness falls significantly below the layer-local population:

$$\widehat{\Phi}_i^{(k)}(T) < \mathbb{E}_j[\widehat{\Phi}_j^{(k)}(T)] - \lambda \operatorname{SD}_j[\widehat{\Phi}_j^{(k)}(T)].$$

All remaining neurons are assigned to the other group. This produces a partition

$$L_k = S_k \cup E_k \cup O_k,$$

where $S_k$, $E_k$, and $O_k$ denote survived, eliminated, and other neurons in layer $L_k$, respectively.

The experimental analysis is organized around the four operational criteria introduced in Definition 3.14. The variation criterion is evaluated by testing whether neuron-level fitness, utility, trajectory length, and entropy exhibit non-degenerate within-layer dispersion. The competition criterion is evaluated through matched ablation: survived, eliminated, other, and random neuron groups are removed under the same group-size constraint, and the resulting changes in loss and accuracy are compared. The selective retention criterion is evaluated by checking whether survived neurons have larger held-out utility than eliminated or random neurons. The non-circular prediction criterion is evaluated by computing survival labels from early trajectory information and testing whether these labels predict later held-out utility.

This protocol is applied across architectures and datasets of increasing complexity. The smaller-scale experiments are used to isolate the emergence of GDP and ablation-based functional competition, while the larger-scale ResNet-50 experiments on Tiny-ImageNet are used to analyze high-dimensional trajectory geometry, layer-wise selective pressure, and the consolidation of survived neurons in deeper representations. Across all settings, the same NDDS pipeline is used: record trajectories, estimate fitness, compute GDP, classify neurons, and validate the resulting groups through ablation and trajectory analysis.

### A.3 Algorithm

Algorithm 1 describes our empirical procedure for measuring trajectory-based Neural Darwinism. During training, we record neuron-level state trajectories over the monitored layers, including parameter, activation, gradient, and information-related statistics. Each neuron is then assigned an empirical fitness score by combining a validation-set ablation utility with combining validation-set ablation utility, sustained trajectory-based adaptation, and informative activity. The weights $\alpha, \beta, \gamma$ are fixed in advance or selected using only non-held-out data. After layer-wise standardization, these scores are aggregated into a Global Darwinian Pressure and used to classify neurons into survived, eliminated, and intermediate groups. Only after the empirical fitness, GDP, and neuron groups have been fixed do we query $\mathcal{D}_{\text{hold}}$ for downstream ablation audits and predictive-validity tests. Thus, $\mathcal{D}_{\text{hold}}$ is not used to define $\widehat{\Phi}$, to tune $\alpha, \beta, \gamma$, or to assign neurons to groups, preserving the non-circular held-out separation.

### A.4 The quantitative analysis of the relative impact of Utility, Trajectory, and Entropy components

To quantify the relative contribution of the Utility, Trajectory, and Entropy terms in NDDS, we conduct component-wise and pairwise ablation studies on ResNet-18 trained on CIFAR-10 under different weight sparsity levels. Table 1 reports the accuracy drop after pruning, where lower values indicate better preservation of the dense model performance. Several observations can be made. First, using any single component alone leads to substantially larger degradation, especially under high sparsity. For example, at 50% sparsity, the single-component variants incur accuracy drops between 27.07% and 28.82%, suggesting that no individual criterion is sufficient to robustly identify dispensable weights. Second, combining two components

---

**Algorithm 1** Empirical NDDS for Trajectory-Based Neural Darwinism

---

**Require:** Network $f_\theta$; monitored layers $\{L_k\}_{k=1}^{D}$; datasets $\mathcal{D}_{\text{train}}, \mathcal{D}_{\text{val}}, \mathcal{D}_{\text{hold}}$; checkpoints $t_0, \ldots, t_T$; weights $\alpha, \beta, \gamma > 0$; threshold $\lambda > 0$.

**Ensure:** Neuron trajectories $\widehat{\Gamma}$, fitness scores $\widehat{\Phi}$, Global Darwinian Pressure $\widehat{\text{GDP}}$, groups $S_k, E_k, O_k$, and held-out audit statistics.

1: Initialize all neuron trajectories $\widehat{\Gamma}_i^{(k)} \leftarrow \emptyset$.
2: **for** $e = 0, \ldots, T$ **do**
3:      Train or update $f_\theta$ on $\mathcal{D}_{\text{train}}$.
4:      Record each monitored neuron state

$$\widehat{\psi}_i^{(k)}(t_e) = [w_i^{(k)}(t_e), b_i^{(k)}(t_e), \mu_i^{(k)}(t_e), g_i^{(k)}(t_e), I_i^{(k)}(t_e)]$$

     and append it to $\widehat{\Gamma}_i^{(k)}$.
5: **end for**
6: **for** each monitored layer $L_k$ **do**
7:      **for** each neuron $i \in L_k$ **do**
8:          Compute trajectory length, information activity, and validation utility:

$$\widehat{A}_i^{(k)} = \sum_{e=1}^{T} \left\| D^{(k)}\left(\widehat{\psi}_i^{(k)}(t_e) - \widehat{\psi}_i^{(k)}(t_{e-1})\right) \right\|_2,$$

$$\widehat{H}_i^{(k)} = \frac{1}{T}\sum_{e=1}^{T} \widehat{I}_i^{(k)}(t_e),$$

$$\widehat{U}_i^{(k),\text{fit}} = \mathbb{E}_{\mathcal{D}_{\text{val}}}\left[\mathcal{L}(f_{\theta \backslash i}(x), y) - \mathcal{L}(f_\theta(x), y)\right].$$

9:      **end for**
10:      Standardize $\widehat{A}^{(k)}$, $\widehat{H}^{(k)}$, and $\widehat{U}^{(k),\text{fit}}$ within $L_k$.
11:      Compute neuron fitness:
$$\widehat{\Phi}_i^{(k)} = \alpha \widehat{U}_i^{(k),\text{fit}} + \beta \widehat{A}_i^{(k)} + \gamma \widehat{H}_i^{(k)}.$$

12: **end for**
13: Compute global pressure:

$$\widehat{\text{GDP}} = \frac{1}{N}\sum_{k=1}^{D}\sum_{i=1}^{n_k} \widehat{\Phi}_i^{(k)}.$$

14: **for** each monitored layer $L_k$ **do**
15:      Partition neurons into

$$S_k = \left\{ i : \widehat{\Phi}_i^{(k)} \geq \mathbb{E}_j[\widehat{\Phi}_j^{(k)}] + \lambda\,\text{SD}_j[\widehat{\Phi}_j^{(k)}], \;\; \widehat{\Phi}_i^{(k)} \geq \widehat{\text{GDP}} \right\},$$

$$E_k = \left\{ i : \widehat{\Phi}_i^{(k)} < \mathbb{E}_j[\widehat{\Phi}_j^{(k)}] - \lambda\,\text{SD}_j[\widehat{\Phi}_j^{(k)}] \right\}, \qquad O_k = L_k \setminus (S_k \cup E_k).$$

16: **end for**
17: Freeze $\widehat{\Phi}$, $\widehat{\text{GDP}}$, and $\{S_k, E_k, O_k\}$.
18: On $\mathcal{D}_{\text{hold}}$, evaluate held-out ablation utilities $\widehat{U}_i^{(k),\text{hold}}$ and compare the effects of $S_k$, $E_k$, $O_k$, and matched random groups $R_k$.
19: **return** $\widehat{\Gamma}$, $\widehat{\Phi}$, $\widehat{\text{GDP}}$, $\{S_k, E_k, O_k\}$, and held-out audit statistics.

---

consistently improves over their corresponding single-component counterparts, demonstrating that these signals are complementary. Among the pairwise variants, $\overline{U}_i^{(k)} + \mathcal{A}_i^{(k)}$ performs best at higher sparsity levels, achieving drops of 8.68% and 20.23% at 40% and 50% sparsity, respectively, while $\overline{U}_i^{(k)} + \overline{\mathfrak{H}}_i^{(k)}$ is more effec-

Table 1: Quantitative analysis of NDDS component-wise and pairwise ablations on ResNet-18 trained on CIFAR-10 under different weight sparsity levels. We report accuracy drop after pruning. **Bold** indicates the best result, and underlining indicates the second-best result.

|  | 20 | 30 | 40 | 50 |
|---|---|---|---|---|
| $\overline{U}_i^{(k)}$ only | 1.88 | 3.04 | 9.01 | 27.07 |
| $\mathcal{A}_i^{(k)}$ only | 2.02 | 3.59 | 10.50 | 28.82 |
| $\overline{\mathfrak{H}}_i^{(k)}$ only | 1.88 | 3.74 | 9.49 | 28.56 |
| $\overline{U}_i^{(k)} + \mathcal{A}_i^{(k)}$ | 1.34 | 2.68 | 8.68 | 20.23 |
| $\overline{U}_i^{(k)} + \overline{\mathfrak{H}}_i^{(k)}$ | 1.31 | 2.22 | 8.84 | 21.49 |
| $\mathcal{A}_i^{(k)} + \overline{\mathfrak{H}}_i^{(k)}$ | 1.36 | 2.98 | 8.75 | 27.02 |
| **Full NDDS** | **0.62** | **1.85** | **6.26** | **12.31** |

tive at lower sparsity levels. Most importantly, the full NDDS formulation consistently achieves the lowest accuracy drop across all sparsity levels, reducing the drop to 0.62%, 1.85%, 6.26%, and 12.31% from 20% to 50% sparsity. The performance gap becomes particularly pronounced in the high-sparsity regime: at 50% sparsity, full NDDS reduces the accuracy drop by 7.92 percentage points compared with the best pairwise variant and by more than 14 percentage points compared with any single-component baseline. These results validate that Utility, Trajectory, and Entropy capture distinct and mutually beneficial aspects of weight importance, and that their joint modeling is crucial for stable pruning under aggressive sparsification.

## A.5 Additional Experiments on Three-layer MLP-Net with MNIST

### A.5.1 Dynamics Neuron Trajectory and Evolution Analysis.

Figure 6(a), top shows the PCA-projected trajectories of shallow-layer neurons across training. Survived neurons (green) follow relatively long and directed paths, indicating sustained representational change. Their motion exhibits fewer reversals than eliminated neurons (red), which instead display short and irregular trajectories, often collapsing toward the origin. This contrast is reflected quantitatively in Figure 6(c), top, where cumulative trajectory length grows steadily for survived neurons. The weight dynamics in Figure 6(d), top reinforce this pattern: survived neurons exhibit increasing $L_2$ norms of incoming weights, whereas eliminated neurons remain almost flat, suggesting a gradual withdrawal of representational capacity. Collectively, these results indicate that even in the shallow layer, gradient descent implicitly differentiates between neurons that maintain sustained alignment with the loss signal and those that do not.

In the middle layer (Figure 6(a), middle), the divergence becomes more pronounced. Survived neurons trace longer and more coherent trajectories, while eliminated neurons remain short and close to the origin. This is supported by Figure 6(c), middle, where the cumulative trajectory length of eliminated neurons grows at a substantially lower rate than that of survived neurons, already showing a marked slowdown by Epoch 2. Weight norms (Figure 6(d), middle) again show a separation, with growth for survived neurons and almost stagnation for eliminated ones. Compared to the shallow layer, the selective bottleneck appears stronger: neurons that fail to establish early alignment with the optimization signal are rapidly marginalized. This suggests that middle-layer neurons, receiving both bottom-up and top-down gradients, undergo more stringent selection toward functional specialization.

The deep layer presents a smaller sample size, but a similar trend is observable. As shown in Figure 6(a), bottom, survived neurons follow more extended trajectories, while the eliminated neuron remains nearly static. Correspondingly, trajectory length (Figure 6(c), bottom) and weight norm evolution (Figure 6(d), bottom) both indicate continued adaptation for survived neurons but not for the eliminated one. Although the limited number of neurons precludes strong statistical claims, the observed divergence suggests that selection pressures persist even near the output. Importantly, this implies that architectural proximity to the loss signal alone does not guarantee survival; functional alignment remains necessary.

Overall, Figure 6 highlights a consistent layer-wise pattern: shallow-layer neurons exhibit the earliest divergence, middle-layer neurons experience intensified selection with clearer separation between survived and eliminated groups, and deep-layer neurons—though fewer—still reflect selective retention. These results support the view that neuron survival is not imposed externally but emerges from the training dynamics, with selection pressures varying in strength across depth.

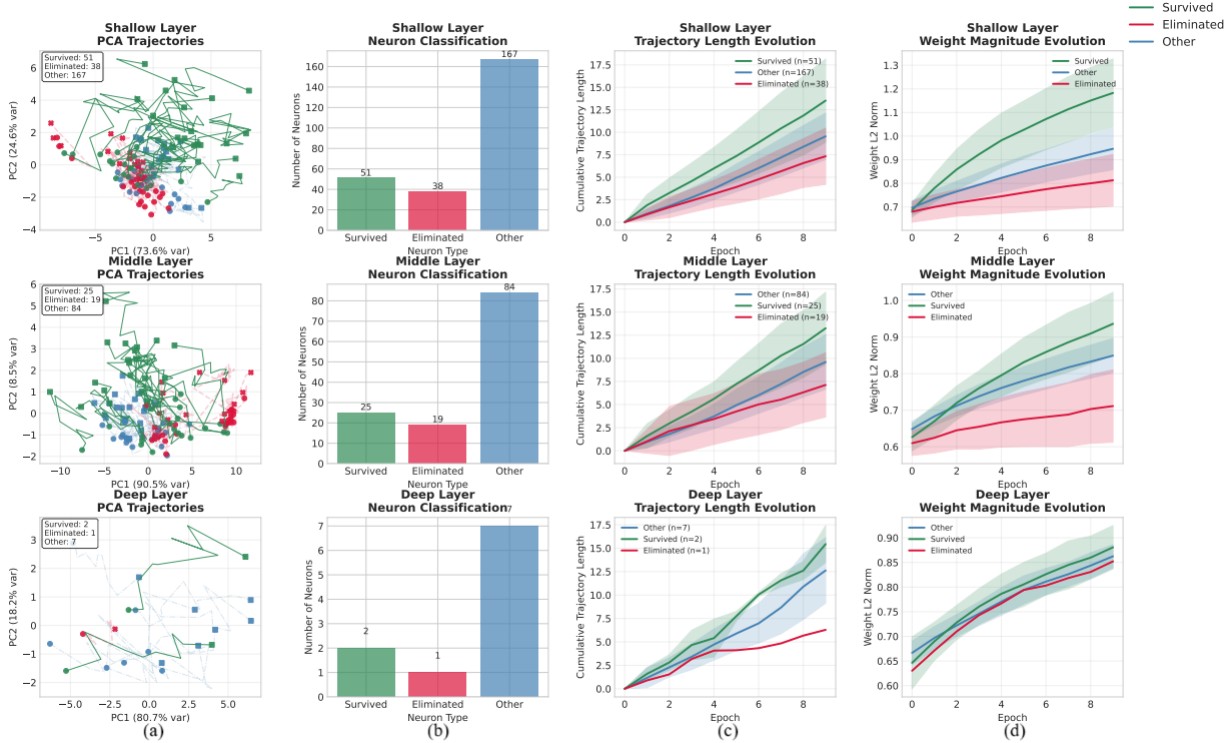

Figure 6: Dynamics Neuron Trajectory and Evolution Analysis on MNIST.

### A.5.2 Static PCA and Global Darwinian Pressure Evolution

Figure 7 (top-left) presents the final-epoch PCA projection of first-layer neuron activations. Neurons categorized as survived occupy relatively dispersed regions, often farther from the origin, which correlates with higher mean Global Darwinian Pressure and greater variance. Eliminated neurons cluster near the origin, suggesting low-pressure states with reduced contribution to the representational space. The majority of neurons fall into the other category, exhibiting intermediate positions without clear clustering, reflecting heterogeneous or drifting roles during training. The Global Darwinian Pressure trajectories (Figure 7, bottom-left) provide a temporal view of this differentiation. Survived neurons increase their average pressure across epochs, indicating sustained engagement with learning signals. Eliminated neurons, in contrast, display a gradual decline toward low, stable pressure levels, consistent with functional silencing. The "other" group remains in an intermediate range, suggesting partial adaptation without clear reinforcement or suppression.

In the middle layer (Figure 7, top-middle), the PCA projection reveals that eliminated neurons are shifted toward the positive-PC1 periphery, while survived neurons occupy a broader and more heterogeneous region spanning both central and peripheral zones. The Global Darwinian Pressure trajectories (bottom-middle) sharpen this divergence: survived neurons exhibit a sustained rise in pressure, whereas eliminated neurons remain suppressed with only marginal growth. Taken as a whole, these patterns suggest that selection-like dynamics manifest most clearly in intermediate layers, where neurons are actively sorted into amplifying versus stagnant trajectories.

For the deep layer (Figure 7, top-right), the neuron count is small (only 2 survived and 1 eliminated), limiting statistical strength. The survived units exhibit higher final mean Global Darwinian Pressure (bottom-right), whereas the eliminated unit declines toward a baseline. While this pattern resembles earlier layers, the small sample size precludes strong generalization.

Overall, the combination of static PCA projections and dynamic Global Darwinian Pressure curves provides complementary evidence of neuron-level differentiation across depth. These results are consistent with the hypothesis that overparameterized networks allocate representational capacity unevenly, with some neurons reinforced while others become marginalized. However, the analyses are correlational and limited by dimensionality reduction and sample imbalance, particularly in deeper layers.

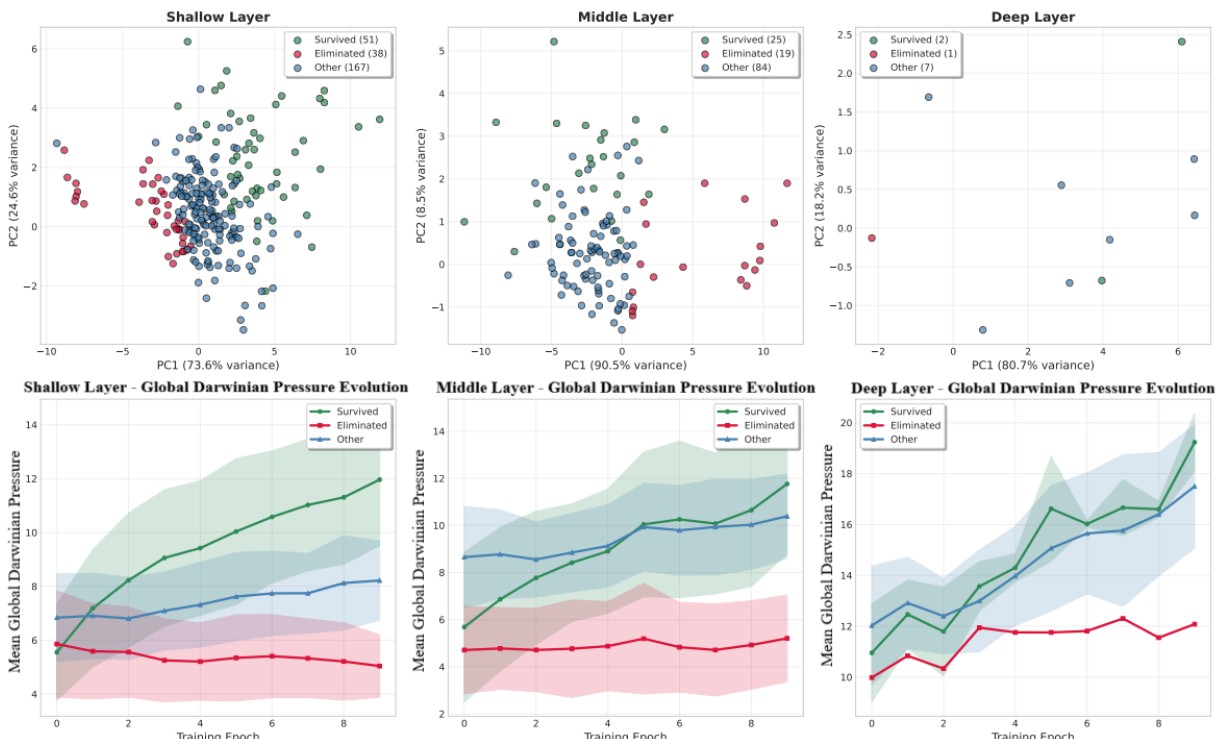

Figure 7: Static PCA and Global Darwinian Pressure Evolution on MNIST.

## A.6 Additional Experiments on ResNet-18 with CIFAR-10

### A.6.1 Dynamics Neuron Trajectory and Evolution Analysis

The shallow layer dynamic PCA trajectories (Figure 8(a), top) show that neuron activations in early convolutional layers—often assumed to encode low-level, generic features—already exhibit signs of representational divergence. Survived neurons tend to follow more stable and moderately directed paths in the PCA manifold, with reduced dispersion over training, suggesting a gradual consolidation toward more compact representational regions. In contrast, eliminated neurons display more irregular trajectories, with frequent directional changes and less coherence, indicating comparatively unstable representational roles. This difference is also reflected in the cumulative trajectory length evolution (Figure 8(c), top): survived neurons maintain consistently higher cumulative movement compared to eliminated neurons, suggesting greater adaptability and sustained representational change across epochs. While the absolute gap is modest, survived neurons display more continuous directional displacement, whereas eliminated neurons tend to plateau earlier, consistent with a potential stagnation of their representational contribution. From a structural perspective, the weight magnitude evolution (Figure 8(d), top) indicates that the convolutional filters corresponding to survived neurons generally retain slightly higher L2 norms throughout training, while those of eliminated neurons

remain lower. This trend is consistent with the interpretation that neurons contributing more strongly to gradient pathways receive relatively greater synaptic reinforcement, whereas others undergo gradual attenuation. Collectively, these results suggest that even shallow layers are subject to competitive dynamics, where only subsets of neurons demonstrating sustained utility remain functionally active.

The middle layers serve as a transitional zone between low-level and high-level representations, and this role is reflected in the diversity of neuron trajectory dynamics. As shown in the dynamic PCA projections (Figure 8(a), middle), neurons in these layers exhibit heterogeneous representational paths over training. Survived neurons tend to follow longer and more coherent trajectories, often traversing distinct regions of the PCA manifold, suggesting a gradual alignment with intermediate-level features. By contrast, many eliminated neurons show less coherent movement, with shorter and more irregular trajectories, though some maintain moderate displacement comparable to the other group. The cumulative trajectory length curves (Figure 8(c), middle) provide quantitative support for these observations: on average, survived neurons reach greater cumulative lengths than eliminated or other neurons, reflecting more sustained representational plasticity. Eliminated neurons continue to grow but at a slower rate, with later signs of stagnation. A similar pattern is visible in the weight magnitude evolution (Figure 8(d), middle), where survived neurons exhibit slightly higher L2 norms than eliminated neurons. Although the difference is modest, its persistence across epochs indicates that neurons contributing more to the task tend to retain larger weight magnitudes. As a whole, these results suggest that the middle layers serve as a representational bottleneck where neurons undergo implicit selection, retaining those with flexible and task-relevant transformations.

In the deep layer, the contrast between neuron groups becomes more pronounced. As illustrated by the dynamic PCA trajectories (Figure 8(a), bottom), survived neurons follow long, smooth, and more aligned paths through representation space, frequently converging to structured low-dimensional subspaces. These neurons appear to encode abstract, class-discriminative information that supports final classification. In contrast, eliminated neurons reveal short, noisy, and non-convergent trajectories, often stagnating or oscillating without clear direction, suggesting limited long-term utility. This distinction is also evident in the trajectory length evolution (Figure 8(c), bottom), where survived neurons maintain the highest cumulative distances relative to eliminated neurons. These lengths reflect sustained representational change that tracks increasing class separability. Moreover, the variance among survived neurons is smaller, suggesting more constrained roles in the deep layer. The weight magnitude evolution (Figure 8(d), bottom) further highlights this separation: survived neurons retain high L2 norms, while eliminated neurons undergo progressive attenuation. The resulting divergence is strongest in this layer, consistent with stronger selective pressure as representations become more task-specific.

Overall, these findings are consistent with the framework of Neural Darwinism: across layers, neurons exhibit competitive dynamics shaped by their sustained utility. While shallow layers already show signs of divergence, the middle layers intensify selective processes, and the deep layers consolidate highly specialized neurons. The evidence from trajectory dynamics and weight evolution collectively supports the interpretation that representational selection operates hierarchically, shaping survival and elimination throughout the network.

### A.6.2 Static PCA and Global Darwinian Pressure Evolution

In Figure 9 left and bottom-left, the PCA projection (97.8% variance explained by PC1) shows that survived neurons occupy a relatively more compact region of the activation space, while eliminated neurons are scattered toward peripheral, low-density zones. Other neurons form a diffuse cloud spanning both regions. The evolution curves of mean Global Darwinian Pressure corroborate this structure: survived neurons sustain moderately higher pressure levels with gradual stabilization, whereas eliminated neurons display persistently weak pressure, and others remain intermediate. These patterns suggest that even at early layers—traditionally considered low-level feature extractors—there is already a degree of representational competition, consistent with the Neural Darwinism view that selection pressure operates from the outset of learning.

In Figure 9 middle and bottom-middle, the PCA embedding (94.2% variance explained by PC1) reveals a clearer differentiation than in shallow layers. Survived neurons cluster more tightly along dominant axes, while eliminated neurons are dispersed across orthogonal or low-density subspaces. Other neurons span an

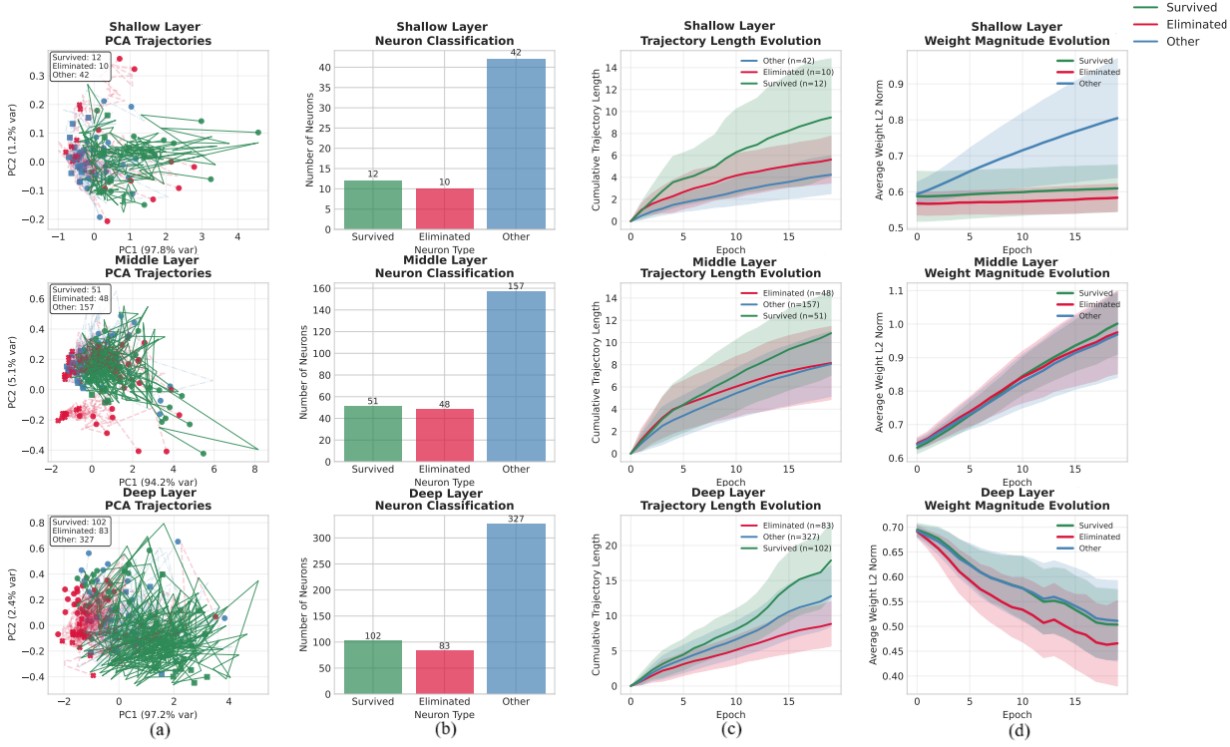

Figure 8: Dynamics Neuron Trajectory and Evolution Analysis on CIFAR-10.

intermediate gradient, partially overlapping both groups. The mean Global Darwinian Pressure dynamics mirror this structure: survived neurons maintain higher, stable pressure levels, whereas eliminated neurons steadily decline. These findings are consistent with the hypothesis that middle layers face stronger selective pressure, as they form an intermediate representational bottleneck where neurons must converge toward task-relevant manifolds to persist.

In Figure 9 right and bottom-right, in the final layer (97.2% variance explained by PC1), survived neurons are broadly distributed along the dominant axis but relatively compact along PC2, indicating alignment to a high-variance representational subspace. Eliminated neurons are concentrated in the lower-PC1 region, while others populate an intermediate zone overlapping both groups. The evolution of mean Global Darwinian Pressure reinforces this separation: survived neurons sustain the highest pressure with relative stability, eliminated neurons remain consistently suppressed, and others occupy intermediate levels. Therefore, the static and dynamic views suggest that deep layers culminate the Darwinian competition, consolidating a high-utility representational manifold surrounded by marginal units.

## A.7 Additional Experiments on VGG-16 on CIFAR-100

### A.7.1 Dynamics Neuron Trajectory and Evolution Analysis

In the shallow layer of Figure 10, the dynamic PCA trajectory analysis reveals early indications of neuronal differentiation consistent with the principles of Neural Darwinism. Survived neurons—characterized by relatively higher activation levels and modestly higher weight magnitudes—tend to originate near the PCA origin at the start of training and progressively diverge along more extended and directionally consistent paths in activation space (Figure 10(a), top). Their trajectories exhibit sustained cumulative displacement over the training epochs (Figure 10(c), top), suggesting continued adaptation. Although the paths are often noisy and irregular, the outward spread indicates a gradual specialization process that may enable distinct low-level feature subspaces to emerge under task-driven gradient signals. By contrast, eliminated neurons generally follow more compact trajectories, remaining closer to the origin and displaying shorter cumulative

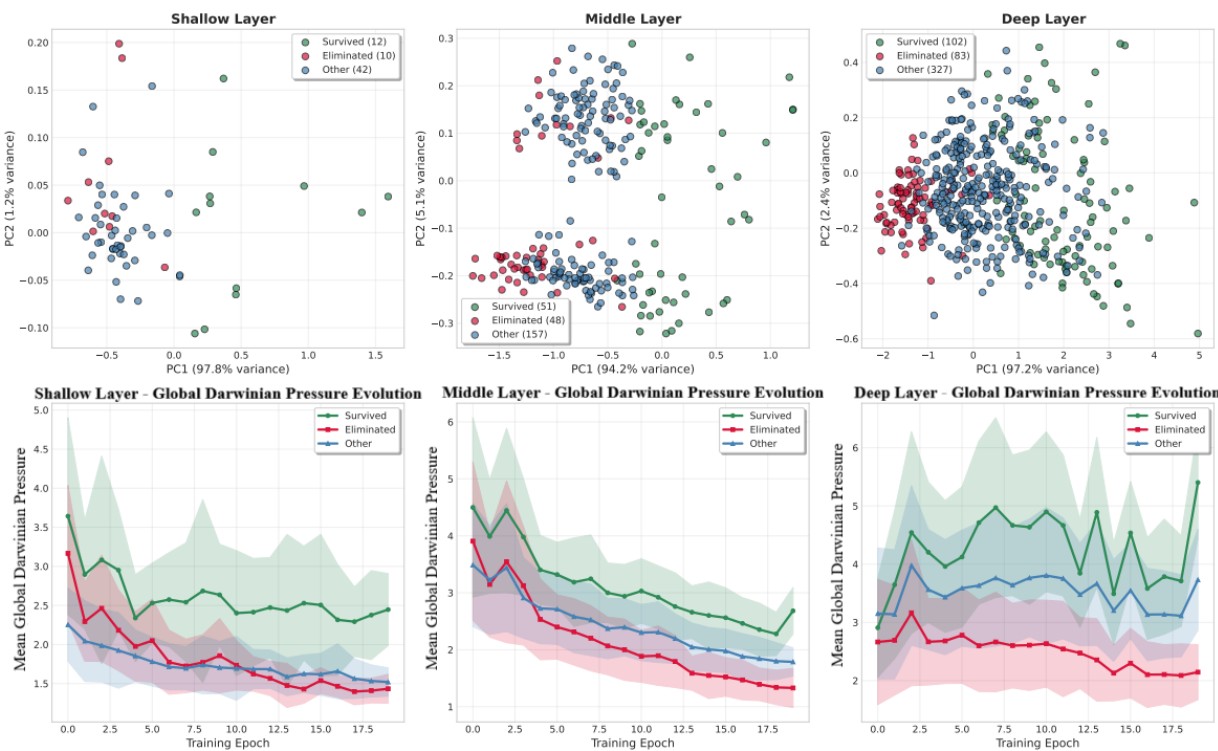

Figure 9: Static PCA and Global Darwinian Pressure Evolution on CIFAR-10.

displacements (Figure 10(a,c), top). Their temporal variance is lower and their trajectory curvature less pronounced, implying reduced representational change. The L2 weight norms of this group are on average slightly lower than those of survived neurons, but the distributions remain strongly overlapping (Figure 10(d), top). While gradient flow is not directly quantified, the limited representational mobility is consistent with the interpretation that these neurons receive weaker or less task-relevant updates during training. The neurons classified as other occupy an intermediate position. Their trajectories are more diffuse and less directionally stable (Figure 10(a), top), with cumulative lengths that are broadly comparable to those of survived neurons but accompanied by larger variance (Figure 10(c), top). Some display periods of outward displacement before stabilizing, while others remain closer to the origin throughout. This heterogeneity suggests that they represent a transitional population whose role is not firmly consolidated within the finite training horizon. Overall, these patterns support a local form of Neural Darwinism: within the shallow layer, a subset of neurons progressively differentiates and maintains higher representational activity, whereas others remain less engaged and gradually lose relative influence. The emergence of such divergence close to the raw input highlights that selection pressures may act from the earliest stages of learning.

In the middle layer—where hierarchical abstractions become more pronounced—the selective dynamics appear intensified relative to the shallow layer. PCA trajectories (Figure 10(a), middle) show that many survived neurons diverge from the origin early and continue outward with sustained displacement, though their paths remain noisy and variable. While most neurons cluster near the PCA origin, a modest subset of survived neurons extends into more distinct regions of the projection space, suggesting partial occupation of differentiated representational subspaces. Eliminated neurons, by contrast, display shorter or less stable trajectories: some show brief excursions before returning toward the origin, whereas others remain in intermediate positions without consistent outward drift. The other neurons again form a heterogeneous group, with some traveling considerable distances but frequently changing direction, and others staying confined near the origin. Quantitatively (Figure 10(c), middle), survived neurons accumulate the greatest trajectory lengths by the final epoch, though the margin over other groups is modest (approximately 0.3–0.4 units). In terms of weight evolution (Figure 10(d), middle), all neuron types exhibit monotonic L2 norm decay,

with survived neurons showing a slightly slower decline and thus ending with marginally higher magnitudes. This suggests that survival is associated with maintaining relatively stronger synaptic weights, though the effect size is small. Collectively, the middle layer illustrates an intensification of competitive dynamics, where survived neurons maintain more persistent representational mobility, eliminated neurons adapt weakly or transiently, and the majority of units remain in flux without converging to stable roles.

In the deep layer—the final fully connected stage before classification—the rate of representational change appears increased, consistent with a late-phase consolidation process. Survived neurons continue to accumulate trajectory length (Figure 10(c), bottom), but at a quicker rate compared to earlier layers. In the PCA projection (Figure 10(a), bottom), these neurons drift outward from the origin and follow moderately directed paths, with curvature and displacement gradually increasing over time. This pattern indicates partial stabilization, consistent with their role in encoding higher-level, semantically richer features that require fewer adjustments once tuned. Weight magnitude curves (Figure 10(d), bottom) similarly show that survived neurons maintain slightly higher norms than eliminated and other neurons, though the separation remains limited. Eliminated neurons in the deep layer exhibit shorter cumulative trajectory lengths and modestly lower weight norms. While some early movement is evident, their displacement growth slows considerably, and their PCA positions remain relatively central, indicating constrained representational change. The other group again occupies an intermediate position, with moderate representational shifts and weight growth, suggesting residual but limited contribution to the final predictive function.

In summary, these observations align with a Neural Darwinism perspective in which neuronal survival reflects continued representational mobility and modestly stronger synaptic weights, while elimination corresponds to reduced or transient adaptation. Importantly, the presence of a large heterogeneous other group underscores that selection pressure operates continuously, and many neurons remain in transition rather than converging to stable roles. The progression from shallow to middle to deep layers reflects a gradual sharpening of selection, culminating in a smaller set of stabilized neurons in the deepest layer.

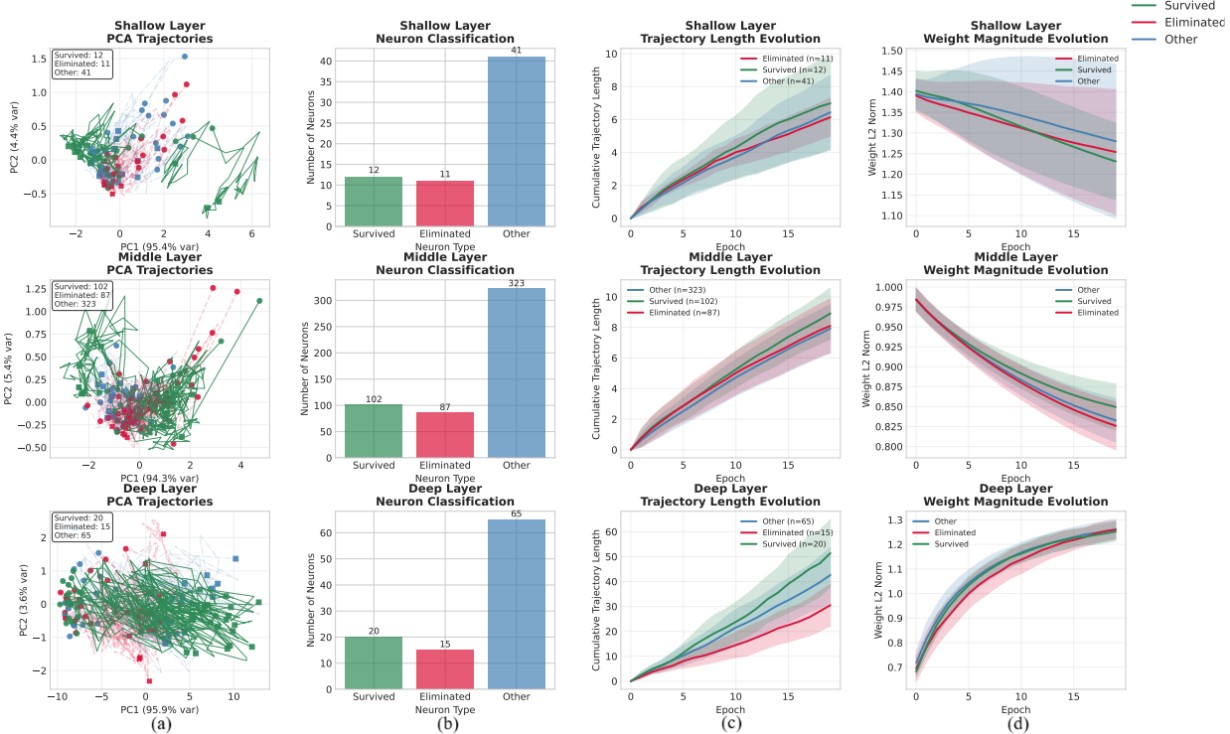

Figure 10: Dynamics Neuron Trajectory and Evolution Analysis on CIFAR-100.

### A.7.2   Static PCA and Global Darwinian Pressure Evolution

In the shallow layer, the final-epoch PCA projection in Figure 11 left shows that the first two principal components account for approximately 99% of the total variance (PC1: 95.4%, PC2: 4.4%), indicating that most inter-neuron activation variability can be represented in a low-dimensional subspace. Despite the limited receptive fields of early convolutional layers, survived neurons (green) occupy more peripheral regions of the PCA plane, with greater dispersion from the origin and from one another, suggesting a tendency toward differentiated feature sensitivities. By contrast, eliminated neurons (red) remain densely concentrated near the origin, reflecting low variance and limited representational differentiation. The Global Darwinian Pressure trajectories in Figure 11 bottom-left reinforce this observation: neurons with persistently higher mean Global Darwinian Pressure tend to survive, while those with steadily declining pressure levels move toward elimination. The distribution of survived neurons suggests diversity in low-level tuning—potentially edges or localized textures—that broadens the expressive basis available for subsequent layers. While the pattern is not definitive, it is qualitatively consistent with a threshold-like competitive process, in line with selection mechanisms hypothesized in Neural Darwinism.

In the middle layer, the PCA projection in Figure 11 middle explains roughly 99% of the variance (PC1: 94.3%, PC2: 5.4%). Here, survived neurons (green) are broadly distributed across the PCA space, often forming multiple partially separated groups, whereas eliminated neurons (red) cluster tightly near the origin. The other group (blue) occupies an intermediate band, positioned between the high-variance survived regions and the low-variance eliminated cluster. Global Darwinian Pressure evolution patterns (Figure 11 bottom-middle) reveal that survived neurons maintain high and relatively stable pressure levels, eliminated neurons exhibit a consistent decline, and others remain at intermediate levels with mild fluctuations. The spread of survived neurons across the PCA space suggests an increasing degree of representational diversification at this stage, corresponding to the formation of mid-level abstractions. The non-random structure—characterized by local coherence within groups and broader separation between groups—indicates systematic partitioning of representational space. The central concentration of eliminated neurons, coupled with their declining pressure, is consistent with redundancy or reduced gradient flow, whereas the transitional behavior of the other group may reflect delayed specialization.

In the deep layer, corresponding to the final fully connected stage, the PCA projection in Figure 11 right shows that the first two principal components explain about 99% of the variance (PC1: 95.9%, PC2: 3.6%). This high concentration of variance suggests a compressed and highly structured representational space, consistent with the role of this layer in integrating features for classification. Survived neurons are predominantly located in peripheral regions of the PCA plane, often grouped into small clusters. The Global Darwinian Pressure trajectories in Figure 11 bottom-right show that survived neurons maintain higher and often increasing mean Global Darwinian Pressure across training epochs, indicating sustained engagement in the final decision space. By contrast, eliminated neurons cluster near the PCA origin and exhibit consistently lower pressure levels and slower growth, suggestive of early functional deactivation. Other neurons occupy intermediate positions, with pressure dynamics reflecting transient or weak selectivity that does not consolidate into either survival or elimination.

Overall, the three-layer comparison in Figure 11 highlights a consistent pattern: variance in activations is concentrated in a few dominant dimensions, survived neurons occupy more dispersed regions and sustain higher Global Darwinian Pressure, while eliminated neurons remain near the origin with declining pressure levels. The other group exhibits transitional characteristics, reflecting instability or incomplete specialization. The combined static and dynamic views are qualitatively consistent with a selection-based process in which functionally distinctive neurons persist and redundant ones fade, echoing principles of Neural Darwinism.

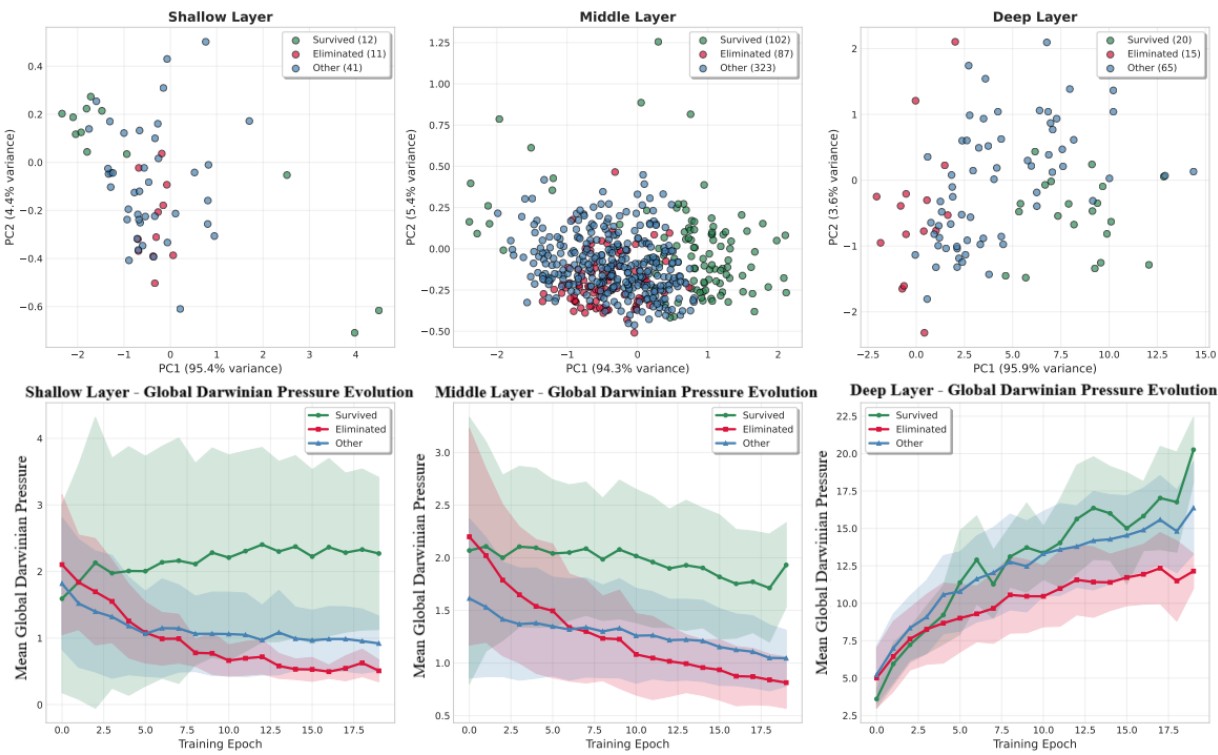

Figure 11: Static PCA and Global Darwinian Pressure Evolution on CIFAR-100.

