# OpenReview forum: "Trajectory-Based Neural Darwinism in Convolutional Neural Networks: Variation, Competition, and Selective Retention"
_TMLR — Rejected by TMLR_

### Review · Reviewer_GqPj · 2026-03-10

**Summary Of Contributions:**

The paper proposes NDDS, a framework to assess the fitness of neurons and groups of neurons in an artificial neural network. NDDS quantifies fitness through a combination of utility (how much the neuron is important to preserving good performance), trajectory length (how much the neuron's representations change over time) and the neuron's entropy. Fitness is then used as criteria to determine whether a neuron should survive or be eliminated. Furthermore, the authors show a relationship between fitness and a quantity known as gradient-variance contribution. Finally, the authors experimentally show a number of results including that NDDS correctly predicts which neurons are important for performance and analyses showing how selective pressures vary over depth and training time.

**Audience:**

Yes

**Audience Explanation:**

Yes: evolutionary dynamics at the neural level is an active, yet underexplored area. More work in this field would certainly be of interest to researchers studying dynamically changing neural architectures and similar topics.

**Broader Impact Concerns:**

No broader impact concerns.

**Claims And Evidence:**

Yes

**Claims Explanation:**

Overall, the theoretical results appear correct and the experimental analysis is also solid- the authors consider a number of datasets and architectures up to a Resnet-50 trained on TinyImagenet. The value of NDDS for analysis of neural networks is made clear.

On the other hand, the design choices of NDDS are not well-justified. For instance, the neuron state vector includes the time-averaged gradient and entropy; why these statistics and not others? Also, why is the fitness a convex combination of utility, trajectory length and entropy- why not just utility? Is there a reason why the elimination criterion is not just the opposite of the survival criterion? There are many choices like this that are at best justified with qualitative statements, but without theoretical rigor. If the purpose of the paper is to introduce a new set of analysis tools for neural networks (as it seems to be), more carefully justifiying these is important.

Relatedly, there are not sufficient ablations for each of the choices made in the design of NDDS. For instance, what happens if the entropy term is ablated from the definition of fitness (I may have missed it, but I don't see an ablation considering this)? Similarly, what if components of the neuron state vector are ablated?

Given that the focus of the paper seems to be presenting a useful set of analysis tools rather than a better criterion for eliminating/pruning neurons (or some other practical goal), the lack of Imagenet scale experiments or more baseline methods is not as much of a concern.

**Requested Changes:**

**Critical for acceptance**
- More carefully justify the design choices of NDDS
- Additional ablations for each of the components of NDDS

**Would strengthen**
- Section 2.2 title: should it be neuron Darwinism?
- An explicit equation defining the entropy proxy in equation 5 would be useful
- Text in figures is too small
- It's unclear what the error margins represent in figures 4 and 5

---

### Review · Reviewer_FpY9 · 2026-03-31

**Summary Of Contributions:**

The authors define an evolutionary fitness function for neurons based on a linear combination of integrated quantities including the total trajectory length of its state vector (which includes its parameters, moments of the activations, and information theoretic quantities), the entropy, and an ablation-based utility function which measures the loss with and without the activation. They also defined a way to use the fitness metric to characterize neurons as high fitness (survived) or low fitness (eliminated).

The authors then conducted experiments using these metrics. They showed that removing neurons with low fitness degraded performance less than removing neurons with high fitness or random fitness in an MNIST experiment. They then show evidence that grouping neurons by fitness results in interpretable PCA trends as a function of layer and training time.

**Audience:**

No

**Audience Explanation:**

As currently written, it is completely unclear what to take away from the definitions and experiments. The fitness functions are so complex as to not be interpretable, and the results themselves do not appear to be so compelling as to convince practitioners to try these very complicated fitness measurements over simpler ideas like norm or influence based metrics for ranking neurons by importance.

**Claims And Evidence:**

No

**Claims Explanation:**

The overall definitions are quite confusing; for example Equation 10 includes a block diagonal rescaling without any reference to why and how the specific rescaling is computed. The definition of the fitness itself includes many terms, some of which are redundant, and has $3$ tunable hyperparameters. In addition, the selection criterion itself has a hyperparameter.

This means that there is some high dimensional combination of statistics with $4$ tunable hyperparameters that is used to stratify the neurons into different groups which are used in the pruning and PCA analyses. It is not clear to me if the corresponding results and analyses are then due to some clever identification of relevant properties of neurons, or if there was so much information stuffed into the fitness that some combination of hyperparameters will reveal statistically significant trends in the experimental settings regardless of what was put into the fitness. It's also not clear how much each of the various terms contribute to the rankings. This is in contrast to other work on e.g. pruning which uses much simpler statistics to stratify neurons, and shows larger effects.

**Requested Changes:**

How does this definition of fitness correlate to other measures used to understand the importance of various neurons? How much do the different components contribute to the fitness? How much does tuning the coefficients of the fitness matter to the results?

---

### Review · Reviewer_wqLN · 2026-04-19

**Summary Of Contributions:**

The paper proposes a framework for finding neurons which contribute more towards performing a certain role in a network than others. This I achieved by defining three quantities: 1) the time averaged utility which measures the impact on the loss from deleting a neuron, averaged over all training steps; 2) the trajectory length which is the path integral of a vector of various parameters and statistics including the average activation and derivative of the neuron over the dataset; 3) the accumulated entropy of a neuron over all training steps. The convex combination of these quantities after normalisation is defined the evolutionary fitness and used to track the fitness of the neuron (in an evolutionary sense). By averaging this quantity over all neurons you get an aggregate statistic for the fitness of the network.

The paper goes on to present experiments which are left relatively vague and claim to support the use of this metric. However, it seems this requires an assumption that the network is initialised with small weights (in the learning regime, as the notion of neuron dynamics depicting importance is incongruent with the lazy regime by definition) which is not mentioned. Further, as I will discuss below, these experiments are not interpreted fairly and they are not sufficiently controlled to fairly assess the utility of the proposed framework.

**Audience:**

No

**Audience Explanation:**

What the paper proposes would be of interest. However, in its current form, it is unlikely that a reader has sufficient information to even reproduce the experiments without concerning about the framework being presented. Thus, the unclear benefit of the framework and lack of experimental control undermines the majority of the paper.

**Broader Impact Concerns:**

The framework presented is theoretical and experiments conducted on small datasets. Thus, I see no need for a broader impact statement or cause for concern.

**Claims And Evidence:**

No

**Claims Explanation:**

Up to section 4 the paper mainly defines the framework and Section 4 provides the evidence to support the framework. Focusing on Section 4 there are a couple concerns:

Firstly, the results are not interpreted fairly. For example in reference to Figure 1 it is said "across all layers we observe a constant shift towards higher GDP" across training. But Figure 1 clearly shows that layer 1 (of a 3 layer network) has a GDP which decreases. Further, no error bars or notion of variance is provided on these figures and so it is unclear how reliable any of Figure 1 is in general. Similarly, Figure 2 and 3 are discussed together in the text as Figure 2 shows what happens to the network when random neurons are pruned, while Figure 3 shows what happens when the fitness metric is used to prune different categories of neurons. However, these are completely incomparable how they are plotted and so almost no insight can be derived from this. The last paragraph of Section 4.2.1 for example claims that the results from the tSNE plots support a Darwinian view but have nothing to do with the fitness function put forward and just show the latent space progressively being corrupted as neurons are deleted. Another example is Figure 4, which shows no statistical significant difference between "survived", "eliminated" or "other neurons" (the three categories reflecting the various importance levels of neurons). The PCA plots are also very impenetrable but have clear contradictions for all layers that the grouping of neurons has any measurable distinctions.

This last point leads to my second and third concern. One is that no explanation is given about how these experiments are conducted beyond stating the architecture name and dataset. The other is that almost no criteria is given for what exactly a Darwinian view of neural dynamics is, which allows the authors to freely claim that most things are Darwinian. Again, the first set of tSNE results do not allude to any of the definitions in the first half of the paper to support their claims. But most concerning is the lack of clarity on what is actually being conducted. The text talks about neurons being "reallocated" after some neurons are pruned, which implies that the network is being retrained after each iteration of pruning. But this would just mean that when the performance drops, the network is likely reaching a capacity in terms of the minimum number of neurons required to do a task. If the paper's only point is that there is excess capacity in a network and that the importance of a neuron depends on how quickly it learns, then this point has been made with far clearer experiments and theories in the past [1,2]. Moreover, the paper does not even make it clear whether the accuracy being measured is training or validation accuracy.

The lack of description of the experiments also hides a lack of controlled experimentation. As the paper claims to show that deleting "surviving" neurons (those which are important) has a greater effect than deleting "Eliminated" neurons (those with less importance). However, no mention is made of how many neurons are actually deleted in each case and without deleting a set number of neurons from each category, I struggle to see how this could possibly support any conclusions about the utility of this distinction. Similarly, returning to the PCA plots - the definition of the neuron groupings is based on the magnitude of the neurons explicitly (particularly through quantities like the trajectory length) but then the magnitude of the neurons in the PCA space is used to support that some distinction exists between the neurons. The lack of detail about what exactly is being done in these plots makes it difficult to be sure, but this seems like circular reasoning.

In summary, the experiments are not described with sufficient clarity, the results depicted clearly contradict the use of the proposed framework anyway and yet these contradicting points are ignored and used to support the framework instead, with statements like "survived neurons follow long, structured arcs" which have no clear correspondence to anything defined or depicted.

[1] Frankle, Jonathan, and Michael Carbin. "The lottery ticket hypothesis: Finding sparse, trainable neural networks." arXiv preprint arXiv:1803.03635 (2018).
[2] Saxe, Andrew, Shagun Sodhani, and Sam Jay Lewallen. "The neural race reduction: Dynamics of abstraction in gated networks." International Conference on Machine Learning. PMLR, 2022.

**Requested Changes:**

The paper needs to do the following:
- Clearly define what exactly is needed to constitute Darwinian dynamics and motivate why this concept provides a helpful perspective
- Explain the experiments in significantly more detail
- Reconsider how it depicts the results and how to appropriately measure the effect of Darwinian dynamics which is not true by construction
- Tie all conclusions back to a defined metric in the framework or some measurable definition of success
- The paper needs to make the distinction between the learning and lazy regimes [3] of network initialisation and show that the framework is applicable to both settings if it is going to put it forward as a general framework for the field.

[3] Geiger, Mario, et al. "Disentangling feature and lazy training in deep neural networks." Journal of Statistical Mechanics: Theory and Experiment 2020.11 (2020): 113301.

---

### Decision · Action_Editor_ajtk · 2026-05-19

**Recommendation:** Reject

**Additional Comments:**

We note that the authors submitted a reply on May 16th. However, this response was submitted after the standard 3-week timeline outlined in TMLR's editorial policies (https://jmlr.org/tmlr/editorial-policies.html). By the time this response was received, the reviewers had already evaluated the manuscript, expressed their preferences, and reached a unanimous consensus regarding the paper's shortcomings. Because of this delay, the reply of the authors has been taken into account as it wasn't evaluated by the reviewers.

**Audience:**

No

**Audience Explanation:**

Although the topic of evolutionary dynamics at the neural level is an active and interesting area, the paper in its current state would not serve the TMLR audience well.
The paper suffers from a lack of reproducibility as experiments are poorly described, and the proposed fitness functions are complex and lack interpretability.

**Claims And Evidence:**

No

**Claims Explanation:**

While the theoretical premise of applying Darwinian dynamics to neural networks is intriguing, the reviewers unanimously found the experimental evidence and methodological rigour insufficient to support the paper's claims.

The primary issues include:
- Unjustified complexity and hyperparameters: the method relies on a complex fitness function with multiple tunable hyperparameters. There were concerns about the lack of justification for design choices such as: why the fitness is a convex combination of utility, trajectory length, and entropy.
- Limited ablation studies make it hard to understand whether the results stem from novel properties or simply the high-dimensional tuning of these hyperparameters.
- The interpretation of the experimental results frequently contradicts the provided figures. For instance, the claim of a "constant shift towards higher GDP" across all layers is contradicted by Figure 1.